# Direct observation of topological magnon polarons in a multiferroic material

Song Bao [1,7], Zhao-Long Gu[1,7], Yanyan Shangguan [1], Zhentao Huang[1], Junbo Liao[1], Xiaoxue Zhao[1], Bo Zhang[1], Zhao-Yang Dong[2], Wei Wang [3], Ryoichi Kajimoto [4], Mitsutaka Nakamura[4], Tom Fennell [5], Shun-Li Yu [1,6] ✉, Jian-Xin Li [1,6] ✉ & Jinsheng Wen [1,6] ✉

Magnon polarons are novel elementary excitations possessing hybrid magnonic and phononic signatures, and are responsible for many exotic spintronic and magnonic phenomena. Despite long-term sustained experimental efforts in chasing for magnon polarons, direct spectroscopic evidence of their existence is hardly observed. Here, we report the direct observation of magnon polarons using neutron spectroscopy on a multiferroic $Fe_2Mo_3O_8$ possessing strong magnon-phonon coupling. Specifically, below the magnetic ordering temperature, a gap opens at the nominal intersection of the original magnon and phonon bands, leading to two separated magnon-polaron bands. Each of the bands undergoes mixing, interconverting and reversing between its magnonic and phononic components. We attribute the formation of magnon polarons to the strong magnon-phonon coupling induced by Dzyaloshinskii-Moriya interaction. Intriguingly, we find that the band-inverted magnon polarons are topologically nontrivial. These results uncover exotic elementary excitations arising from the magnon-phonon coupling, and offer a new route to topological states by considering hybridizations between different types of fundamental excitations.

Magnons and phonons, quanta of spin waves and lattice vibrations respectively, constitute two fundamental collective excitations in ordered magnets. When there is a strong magnon-phonon coupling, they can be hybridized to form a gap at the intersection of the original magnon and phonon bands (Fig. 1a)[1–3]. The hybridized bands feature mixed, interconverted and reversed magnonic and phononic characters, and the associated quasiparticles are defined as magnon polarons[1–3]. Magnon polarons can resonantly enhance the spin-pumping effect[4], and provide a phonon-involved way to generate and manipulate spin currents carried by magnons thanks to their hybrid nature[5,6], signifying promising potentials in spintronics

technology[7,8]. More recently, it has been predicted that magnon-polaron bands can exhibit nonzero Chern numbers and large Berry curvatures, giving rise to the thermal Hall effect[9–15].

These proposals have motivated sustained experimental efforts in chasing for magnon polarons[4–6,16–26]. However, direct spectroscopic evidence with their delicate band structures being explicitly unveiled by neutron spectroscopy is still rare. This is primarily because: i) materials with strong magnon-phonon coupling that can result in such excitations with prominent features are scarce; ii) magnons and phonons are rarely observed in the same energy-momentum window by neutron spectroscopy due to their different dynamical structure

---

[1]National Laboratory of Solid State Microstructures and Department of Physics, Nanjing University, Nanjing 210093, China. [2]Department of Applied Physics, Nanjing University of Science and Technology, Nanjing 210094, China. [3]School of Science, Nanjing University of Posts and Telecommunications, Nanjing 210023, China. [4]J-PARC Center, Japan Atomic Energy Agency (JAEA), Tokai, Ibaraki 319-1195, Japan. [5]Laboratory for Neutron Scattering and Imaging, Paul Scherrer Institute (PSI), CH-5232 Villigen, Switzerland. [6]Collaborative Innovation Center of Advanced Microstructures, Nanjing University, Nanjing 210093, China. [7]These authors contributed equally: Song Bao, Zhao-Long Gu. ✉e-mail: slyu@nju.edu.cn; jxli@nju.edu.cn; jwen@nju.edu.cn

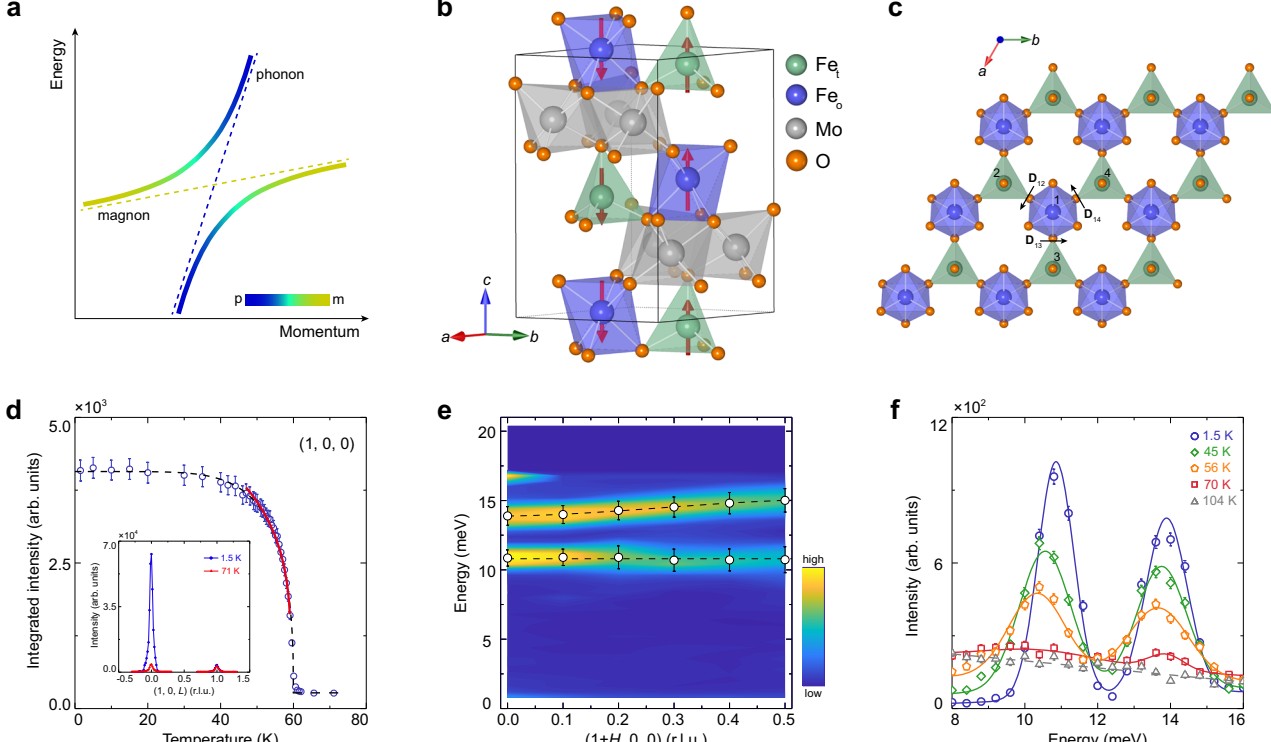

**Fig. 1 | Schematic of magnon polarons and two-dimensional magnetism in a multiferroic $Fe_2Mo_3O_8$. a** Schematic of the magnon-polaron bands after the hybridization between the original magnon and phonon bands illustrated by the thin dashed lines. Colors indicate the weights of the magnonic and phononic components. **b** Crystal and magnetic structures of $Fe_2Mo_3O_8$. The material belongs to the polar $P6_3mc$ space group (No. 186) and exhibits a collinear antiferromagnetic configuration. $Fe_t$ and $Fe_o$ label the ions in the centers of the tetrahedra and octahedra, respectively. Arrows indicate the magnetic moments on Fe atoms. **c** Top view of one Fe-O layer. Black arrows mark the in-plane DM vectors for the NN sites. **d** Temperature dependence of the integrated intensities of the magnetic Bragg peak (1, 0, 0). Error bars represent one standard deviation throughout the paper. The solid curve is a fit to the intensity with $I \propto (1 - T/T_N)^{2\beta}$, where $T_N = 59.33 \pm 0.07$ K and $\beta = 0.129 \pm 0.005$ are the Néel temperature and critical exponent, respectively. The inset shows elastic scans across (1, 0, $L$) along [001] direction at 1.5 and 71 K. **e** Spin excitation spectra around (1, 0, 0) along [100] direction at 1.5 K. The cut-off bright spot near 17 meV is a false signal caused by the small scattering angle. **f** Energy scans at (1, 0, 0) measured at various temperatures. Solid curves are fits with Gaussian functions. Dashed lines in **d**–**f** are guides to the eye.

factors, which hinders the exploration of the interaction effects between them. To overcome these difficulties, $Fe_2Mo_3O_8$ is a prime candidate material[10,27–33].

$Fe_2Mo_3O_8$ is a multiferroic material with a polar $P6_3mc$ space group (No. 186). Below the Néel temperature $T_N \sim 60$ K, both the space-inversion and time-reversal symmetries are broken[27–29,34–36], as shown in Fig. 1b. The magnetism in $Fe_2Mo_3O_8$ arises from $Fe^{2+}$ ions, while $Mo^{4+}$ ions form nonmagnetic spin-singlet trimers[28,29]. The $Fe^{2+}$ ions on each Fe-O layer form a bipartite honeycomb network with different magnetic moments in corner-shared tetrahedra and octahedra (Fig. 1b, c)[34–36]. The absence of an inversion center between nearest-neighbor (NN) Fe sites allows for a non-zero in-plane Dzyaloshinskii-Moriya (DM) interaction. $Fe_2Mo_3O_8$ exhibits a long-range collinear antiferromagnetic order below $T_N$, with antiparallel yet uncompensated moments on each Fe-O layer stacking antiferromagnetically along the $c$ axis (Fig. 1b)[34–36]. Furthermore, it is noteworthy that the magnetic configuration of $Fe_2Mo_3O_8$ can be controlled by either an external magnetic field or chemical doping, leading to a metamagnetic transition into a ferrimagnetic state (Supplementary Fig. 1c, d)[28,29]. More crucially, there has been accumulating evidence that the spin and lattice degrees of freedom are strongly coupled in $Fe_2Mo_3O_8$[10,27–33], rendering it a promising platform to probe the long-sought magnon polarons[4–6,16–26].

In this work, we perform high-resolution neutron spectroscopy measurements and fully map out the magnon and phonon bands in a multiferroic $Fe_2Mo_3O_8$ with strong magnon-phonon coupling[10,27–33]. Due to the acquisition of spin components from magnon conversion,

the acoustic phonons show up together with magnons at small momenta. By examining the interaction effects between them, we directly observe the long-sought magnon polarons, which we show to be topologically nontrivial. These results not only unambiguously identify a new type of excitations, but also provide a fresh ground to study topological states.

## Results

### Magnons at high energies

Figure 1d shows the elastic neutron scattering results for single crystals of $Fe_2Mo_3O_8$. At 1.5 K, a significant increase in the magnetic scattering intensity is observed at the magnetic Bragg peak (1, 0, 0), indicating the establishment of an antiferromagnetic order. On the other hand, (1, 0, 1) corresponds to the magnetic Bragg peak for the ferrimagnetic order[34,36]. The fitting of the temperature dependence of the integrated intensities at (1, 0, 0) yields $T_N = 59.3$ K and a critical exponent of 0.129, which closely matches the expected value of 0.125 for a two-dimensional Ising system[37]. These results are consistent with the magnetic susceptibility measurements (Supplementary Fig. 1c), suggesting that $Fe_2Mo_3O_8$ is a two-dimensional collinear antiferromagnet with strong magnetocrystalline anisotropy along $c$ axis. Figure 1e shows the inelastic neutron scattering (INS) results along [100] direction at 1.5 K, obtained on a triple-axis spectrometer. No acoustic bands are found to disperse up from the magnetic Bragg peak (1, 0, 0). Instead, only two seemingly flat bands with a sizable gap are observed between 10 and 16 meV. Upon warming, these two modes soften slightly, while the scattering intensities decrease dramatically (Fig. 1f).

Above $T_N$, the peaks disappear. This suggests that the two modes correspond to the spin-wave excitations originating from long-range magnetic order, rather than the crystal-electric-field excitations[38]. Given that there are four sublattices of $Fe^{2+}$ ions in the magnetic unit cell of $Fe_2Mo_3O_8$ (Fig. 1b), we consider these two magnon modes to be doubly degenerate in the collinear antiferromagnetic state, as also found in its sister compound $Co_2Mo_3O_8$[39]. We note that the energy scales of the excitation spectra in the other isostructural compound $Ni_2Mo_3O_8$[38] are rather different from those in $Fe_2Mo_3O_8$ (Fig. 1e) and $Co_2Mo_3O_8$[39]. In $Ni_2Mo_3O_8$, the ground state of the crystal-electric-field levels of $Ni^{2+}$ ions is a nonmagnetic singlet, while the interplay between crystal-electric-field effect and magnetic exchange interactions gives rise to a magnetic order with a relatively low transition temperature of $T_N = 5.5$ K. As a consequence, spin waves are observed below 1.5 meV, and higher-energy excitations are believed to arise from the crystal-electric-field spin excitons with a robust temperature dependence[38].

## Anomalous phonons at low energies

To examine the excitation spectra of $Fe_2Mo_3O_8$ in a larger momentum-energy space, we next performed INS measurements on a time-of-flight spectrometer. The two magnon modes between 10 and 16 meV can also be observed at 6 K along both [100] and [110] directions (Fig. 2a, b). These two modes are flat along [001] direction (Fig. 2c), indicating negligible interlayer coupling consistent with the two-dimensional nature of the magnetism deduced from the critical exponent (Fig. 1d). This behavior allows us to integrate the intensities along [001] to improve the statistics when studying the magnon-related excitations, as we did in Fig. 2a, b. Turning to the lower energy

window, we observe additional excitations alongside the two intense magnon modes (Fig. 2a–c). These low-energy modes introduce multiple bands that cannot be solely explained by linear spin-wave theory. Even when accounting for the nonlinearity of spin waves, the extended spin-wave theory fails to account for the presence of multiple bands at significantly lower energies[40]. Therefore, we propose that these additional modes to have a phononic origin, which can also be supported by the following facts. The scattering intensities of these low-energy excitations become stronger as the wavevector **Q** increases (Supplementary Fig. 2a–c), which is characteristic of phonons. Figure 3a shows a specific mode among these excitations, with an onset energy around 5 meV along the [100] direction. Notably, we find such onset excitations at (1, 0, 1) also correspond to the maximum of the excitations along the [001] direction as shown in Fig. 3b, while no significant signal can be observed at the intense magnetic Bragg peak (1, 0, 0) (Fig. 1d). Furthermore, the scattering intensities of such excitations tend to be stronger at larger **Q**s as shown in Fig. 3c. These findings indicate that such a specific mode can be interpreted as phonons connected by a saddle point around 5 meV, rather than gapped spin waves. Our theoretical calculations, which reproduce the multiple bands well, provide further support for the interpretation of these low-energy modes as phonons (Supplementary Fig. 4).

On the other hand, these modes exhibit some anomalous behaviors that suggest they are not ordinary phonons. At 100 K, which is above $T_N$, they become invisible at small **Q**s but persist at large **Q**s, as shown in Fig. 3d–f and Supplementary Fig. 2d–f. Furthermore, it is noteworthy that the saddle point of the phonon spectra around 5 meV, previously considered as an electromagnon (~1.2 THz) in multiferroics

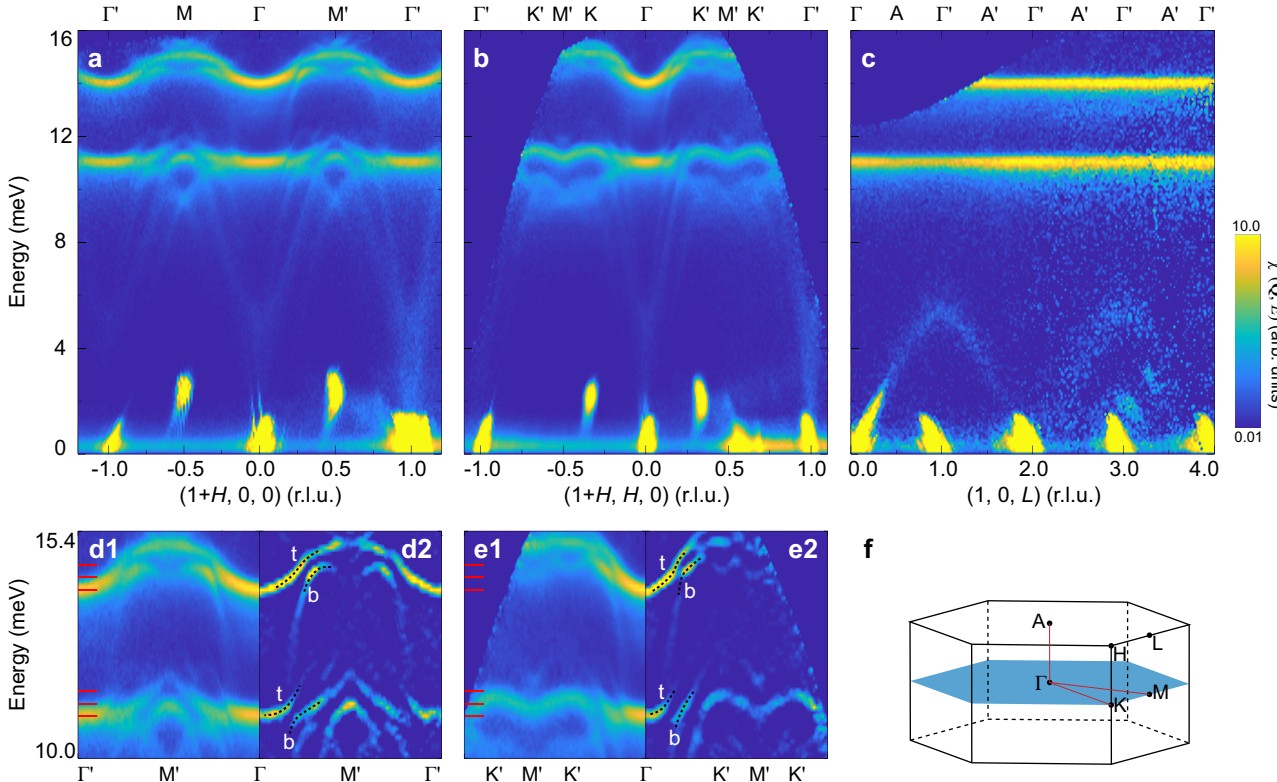

**Fig. 2 | Magnon-polaron excitation spectra. a–c** INS results of the excitation spectra measured at 6 K along [100], [110] and [001] directions, respectively. Where applicable, the integration range used to plot the spectra throughout the paper can be found in Supplementary Table 1. In **c**, raw data on the negative $L$ side have been symmetrised to the positive side to improve the statistics. Due to the limited data coverage, this symmetrisation procedure is applicable only for $L \leq 2$ r.l.u. For better display, these data in **a–c** are plotted in a logarithmic scale of intensity. Bright spots

around 2 meV and elongated signal near the elastic line in **a–c** are spurious peaks caused by saturation of the neutron detectors. **d1, e1** Zoom-in view of magnon polarons in **a** and **b**, respectively. **d2, e2** Spectra obtained by the two-dimensional-curvature method on **d1** and **e1**, respectively. **f** Three-dimensional Brillouin zones with high-symmetry points and paths. Ticks in **d1, e1** mark the energies corresponding to the constant-energy contours plotted in Fig. 4a–f. Dashed lines in **d2, e2** are guides to the eye to mark the top (t) and bottom (b) magnon-polaron bands.

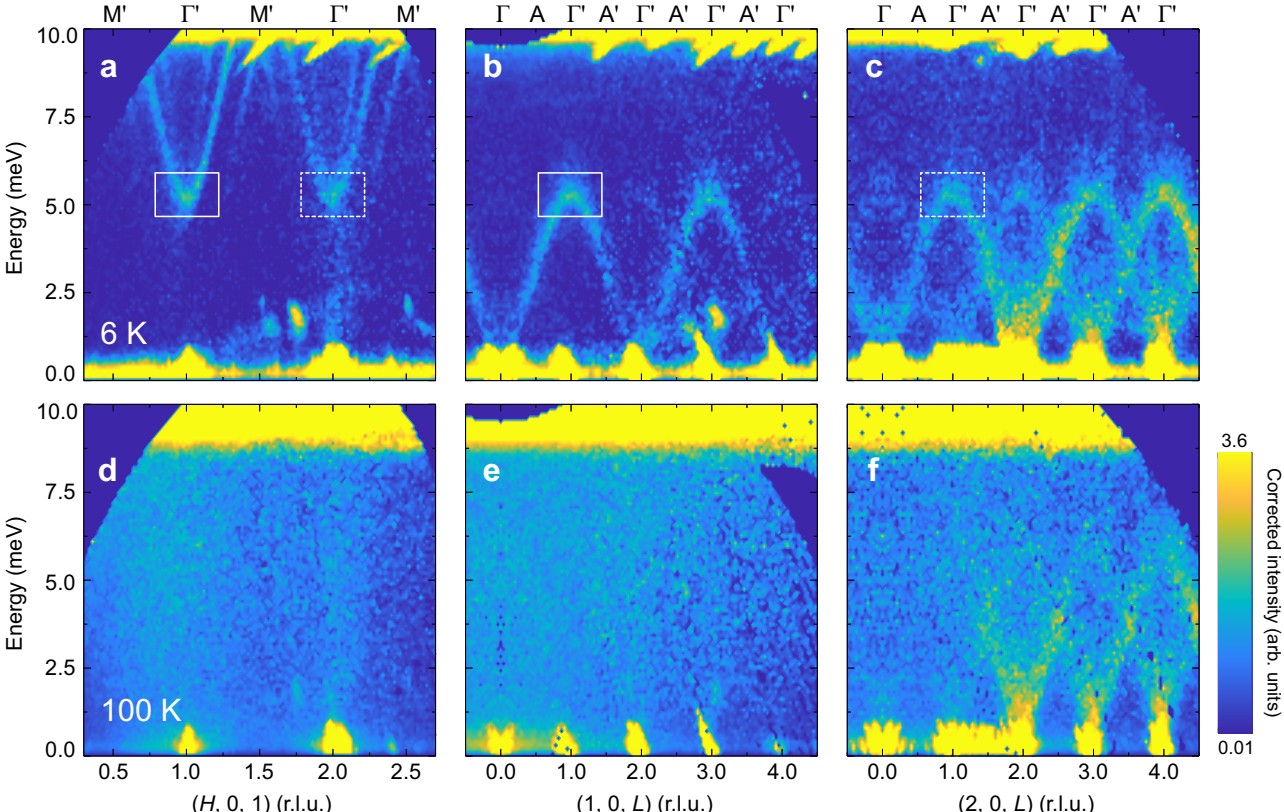

**Fig. 3 | Low-energy phonon excitation spectra.** INS results of the excitation spectra measured at 6 K along [100] (**a**) and [001] (**b**, **c**) directions, respectively. The spectra in **b**, **c** correspond to the out-of-plane variations with different wavevectors $H$s. Solid and dashed squares in **a**–**c** represent the saddle point of phonon spectra around 5 meV at $H = 1$ and 2 r.l.u., respectively. **d**–**e** Same as in **a**–**c**, but measured at 100 K. In **b**, **c**, **e** and **f**, data on the negative $L$ side have been symmetrised to the positive side to improve the statistics. The raw data obtained with lower $E_i = 12$ meV have been corrected by Bose factor and are plotted in a logarithmic scale of intensity.

by some optical measurements[30,31], will be electric-dipole active in the antiferromagnetic phase[30,31]. These results indicate these low-energy phonons acquire some spin components[17] through the strong magnon-phonon coupling, as discussed later, leading to additional magnetic scattering intensities at small $Q$s at 6 K. However, at 100 K, as the magnons collapse, phonons recover their original properties and can only be observed at large $Q$s, where the intrinsic dynamic structure factors are sufficiently large.

**Formation of magnon polarons**

The appearance of acoustic phonons at small momenta together with magnons enables us to examine the interaction effects between them, and we now focus on the regions where the weak and dispersive acoustic phonons tend to intersect with the intense and relatively flat magnons (Fig. 2a, b). Apparently, there exist spectral discontinuities at the nominal intersections between the phonons and magnons (Fig. 2d1, e1), indicative of the gap opening. To better resolve the gaps, we use a two-dimensional curvature method[41] to increase the sharpness of the bands (Fig. 2d2, e2). There are multiple gaps around the two magnon bands at about 11 and 14 meV. Together with the gap opening, the sudden change of the dispersions and the intensities in proximity to the gap unambiguously indicate that magnons and phonons are strongly hybridized and inverted, and consequently two magnon-polaron bands separated by the gap form, as illustrated in Fig. 1a. For the top magnon-polaron band, it changes from dominant magnonic to phononic character as it propagates. As a result, the band's velocity increases but the intensity decreases dramatically around the original intersection. The trend is opposite for the bottom magnon-polaron band. The observation that the low-energy dispersive phonons convert

into magnons with enhanced intensity supports our earlier explanations for the anomalous phonon behaviors, as phonons involved with magnon conversion can carry spins[17]. Importantly, when the magnons collapse above $T_N$, phonons revert to their original dispersions and scattering intensities (Fig. 3 and Supplementary Fig. 2). The spectroscopic characteristics of hybrid magnon polarons and low-energy phonons acquiring spin components in the resonant and off-resonant regions, respectively, are both manifestations of the strong magnon-phonon coupling in $Fe_2Mo_3O_8$.

To further illustrate the formation and evolution of the magnon-polaron excitations, we plot a series of constant-energy ($E$) contours near the two anticrossing regions as shown in Fig. 4a–c (higher energies), and d–f (lower energies). In each region, there are always two sets of excitations around the zone center, representing two magnon-polaron bands separated by the gap. The shapes and intensities of these two excitations change as the energy increases, manifesting the magnon-phonon interconversion within each magnon-polaron band during propagation. To better clarify these, we first plot a set of constant-$E$ scans along [100] direction through the zone center (Fig. 4i), which correspond to the lower anticrossing region (Fig. 4d–f). As the energy increases, it can be found that there is a spectral weight transfer between the two magnon-polaron bands as they propagate. A similar approach can be applied to the constant-$E$ scans at other energies, so that we can extract a series of fitted peak centers and areas. We plot the relative peak position ($\Delta q$) and the ratio of the spectral weight for each band to the total spectral weight of the two magnon-polaron bands as a function of energy (Fig. 4j). When the top (inner) and bottom (outer) magnon-polaron bands are approaching the anticrossing point from low energies, they are of primarily magnonic and phononic natures

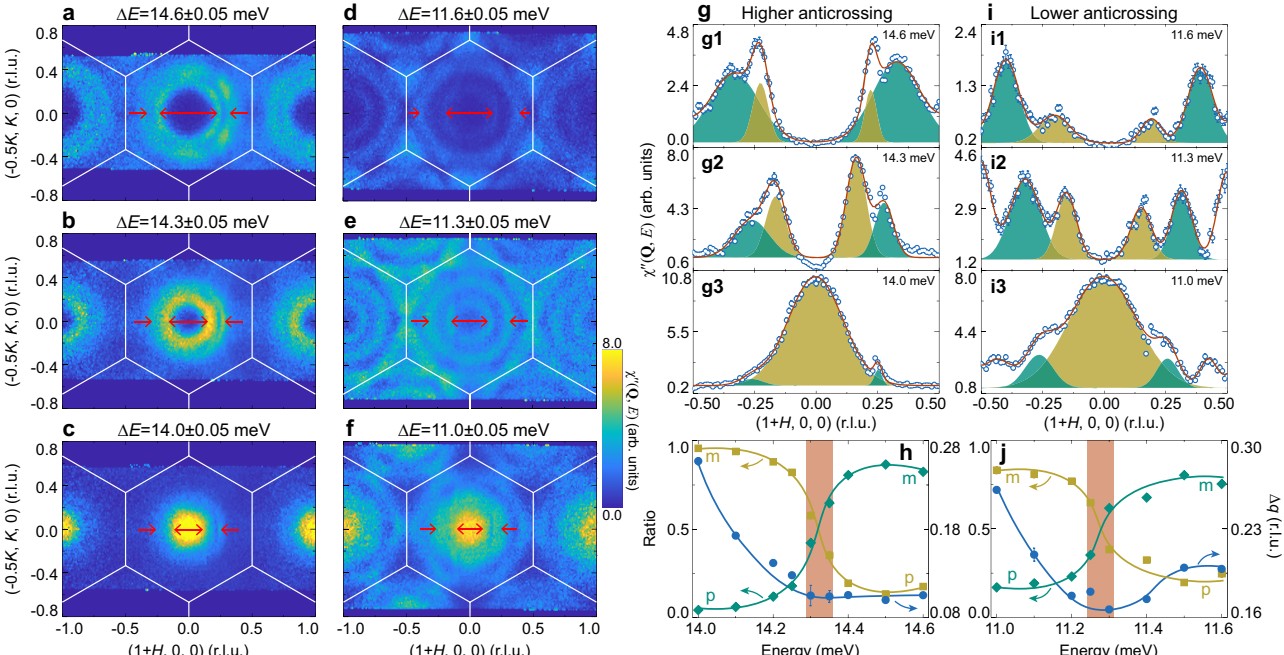

**Fig. 4 | Evolution of the magnon-polaron bands.** Constant-energy contours at a series of selected energies near the higher anticrossing point around 14.3 meV (**a**–**c**) and lower anticrossing point around 11.3 meV (**d**–**f**) in the $(H, K, 0)$ plane. The energies are marked in Fig. 2d1, e1. Arrows mark the two magnon-polaron bands. The inner and outer bands correspond to the top and bottom bands respectively. Solid lines indicate the Brillouin zone boundary. **g, i** Panels labeled 1-3 correspond to the profiles obtained from the cuts in the panels of **a**–**c** or **d**–**f**. Solid curves are fits with Gaussian functions. The yellow and green fillings illustrate the fitted peaks for the two magnon-polaron bands. The fitted areas are labeled as $S_{\mathrm{m}\to\mathrm{p}}$ (yellow) or $S_{\mathrm{p}\to\mathrm{m}}$ (green), based on the direction of the conversion between the magnonic (m) and phononic (p) natures. The asymmetry in the peak shapes on both sides of the origin in **g, i** is caused by the resolution effect on 4SEASONS. **h, j** Ratio of the spectral weight (left axis) and the peak interval ($\Delta q$, right axis) as a function of energy for the higher and lower anticrossing regions, respectively. The ratio is defined as $S_{\mathrm{m(p)}\to\mathrm{p(m)}}/(S_{\mathrm{m}\to\mathrm{p}} + S_{\mathrm{p}\to\mathrm{m}})$ counting spectral weights on both sides of $(1, 0, 0)$. $\Delta q$ is the distance between the peak centers, averaged with the values on both sides. Vertical bars indicate the minimum of $\Delta q$. Solid curves are guides to the eye.

respectively, so the top band dominates the spectral weight as magnons are much stronger. When the two bands reach the anticrossing point, where $\Delta q$ has its minimum, strongest hybridization happens so that the magnonic and phononic components become comparable. Across this position, the main components of the two bands are reversed, and so are their relative intensity ratios. Eventually, the bottom band which is of primarily magnonic nature dominates the spectral weight. Figure 4g, h shows similar behaviors for the constant-$E$ scans corresponding to the higher anticrossing region (Fig. 4a–c). We also examine the magnon-polaron excitations along the orthogonal [−120] direction, and the results are also similar (Supplementary Fig. 3). These results elaborate the band inversion between the original magnon and phonon bands. Together with the gaps in the dispersions (Fig. 2), these constitute the hallmarks for the magnon polarons.

### Dzyaloshinskii–Moriya mechanism and topology

To understand the underlying mechanism of the observed magnon polarons, we develop an effective two-dimensional model that contains both the magnon and phonon terms (Methods). Without the magnon-phonon coupling, the obtained magnon and phonon dispersions are shown in Fig. 5a. Such dispersions capture the main features of the INS spectra (Fig. 2) except for the anticrossings between magnons and phonons. In principle, since magnetic exchange interactions depend on the relative ion positions, magnon-phonon coupling can arise due to the lattice vibrations. In $Fe_2Mo_3O_8$, the magnon-phonon coupling induced by the DM interaction dominates those by other Heisenberg interactions (See details in the Methods). Generally speaking, only the component of the DM vector parallel to the magnetic moments will enter into the spin waves[42]. On the other hand, the DM vector here with only the in-plane component allowed, which is perpendicular to the magnetic moments (Fig. 1b, c), typically will not

contribute to the magnetic ground state as well as the spin excitation spectra[9–11]. We consider it is the lattice vibrations that make the in-plane DM interaction come into play, which in return closely couples magnons and phonons (Methods). The mechanism that the vibrant DM vectors can disturb the procession of the magnetic moments is illustrated in Fig. 5c. In Fig. 5b, the calculated spectra of the coupled system are shown (Parameters can be found in Supplementary Table 2). The gaps between the magnon and phonon bands and interconversions of their spectral components (Fig. 5b) well reproduce the characteristics of the magnon polarons observed experimentally, indicating that the DM-interaction-induced magnon-phonon coupling can give rise to the magnon-polaron excitations. Importantly, near the gaps, it is found that large Berry curvatures can be induced (Fig. 5d, e). Accordingly, the Chern number for each band is calculated and labeled in Fig. 5b. These results show that the magnon-polaron excitations are topologically nontrivial, in accordance with our experimental observation of the band inversion between magnons and phonons. To account for the three-dimensional nature of the phonons, we have also developed a three-dimensional model, as presented in Supplementary Fig. 4.

## Discussion

By now, combing our experimental spectra and theoretical calculations, we have provided compelling evidence that in $Fe_2Mo_3O_8$ there exist topological magnon polarons. Remarkably, in addition to the gap opening between the original magnon and phonon bands, we have also observed the band inversion. Such a case is analogous to that in topological insulators induced by the spin-orbit coupling, but involves the interconversion between magnons and phonons induced by the DM interaction. Therefore, our work provides new perspectives in seeking for topological states—that is to go beyond a single type of

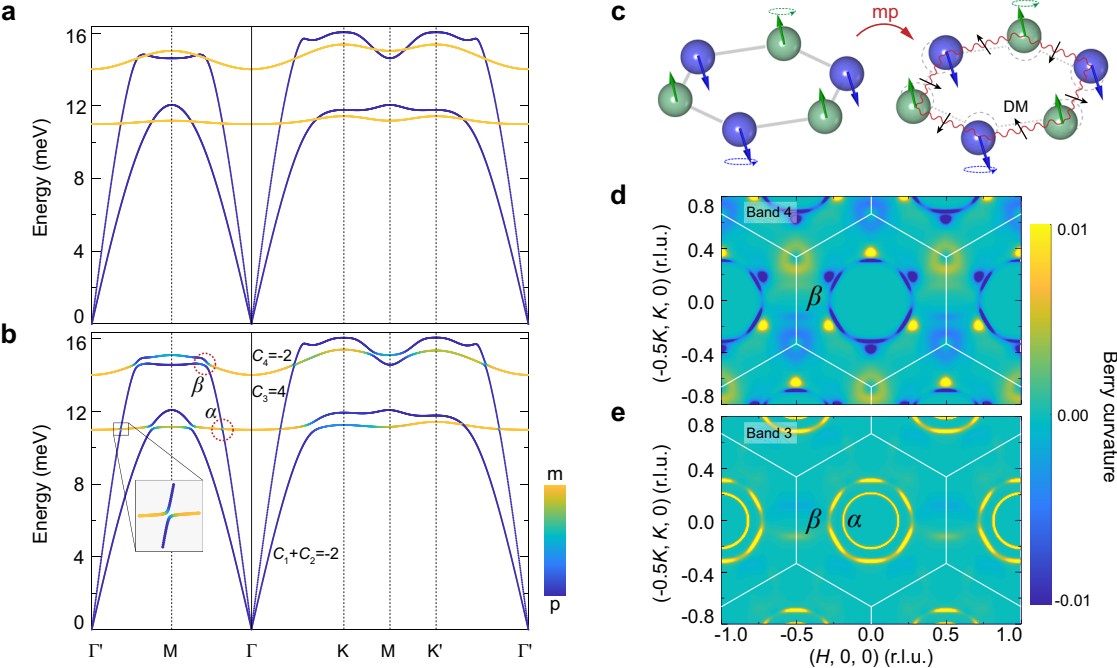

**Fig. 5 | Calculations of the magnon-polaron bands and band topology.**
**a** Uncoupled magnons (orange) and phonons (blue) along the high-symmetry directions without the DM interaction. **b** Topological magnon polarons resulting from the DM-interaction-induced magnon-phonon coupling. The color illustrates the weight of the magnonic and phononic components. Chern numbers from the lowest to highest bands (labeled from 1 to 4) are $C_1 + C_2 = -2$, $C_3 = 4$ and $C_4 = -2$. Here, due to the degeneracy of the acoustic phonons at the $\Gamma$ point, only the total

Chern number is well defined for the lowest two bands. The dotted circles labeled $\alpha$ and $\beta$ correspond to the lower and higher anticrossing regions respectively. **c** Schematic diagram for the DM-interaction-induced magnon-phonon coupling in $Fe_2Mo_3O_8$. Due to the in-plane lattice vibrations, the otherwise canceled DM interaction becomes nonzero and comes into play, which induces the magnon-phonon coupling, labeled as mp. **d**, **e** Calculated Berry curvatures for bands 4 and 3, respectively.

elementary excitations such as electrons, magnons or phonons only, and to consider the topology in their hybrid form.

We note that a recent magneto-Raman scattering study[43] has reported the existence of topological magnon polarons due to the zigzag antiferromagnetic order in the monolayer $FePSe_3$, where the magnon-phonon coupling originates from the anisotropic exchange interactions[43]. It implies that the topological nature of band-inverted magnon polarons can be inherent and resilient, irrespective of the particular form of magnon-phonon coupling[9–15]. In our model calculations, we find that the DM term is larger than the Heisenberg terms (Supplementary Table 2), supporting the strong magnon-phonon coupling induced by the DM interaction in $Fe_2Mo_3O_8$. Notably, this coupling mechanism involves the interaction between in-plane phonons and in-plane magnons that emerges as the leading order in $Fe_2Mo_3O_8$, surpassing the general magnetoelastic coupling[1,13,14] that arises from perpendicular easy-axis anisotropy (See details in the Methods).

The strong DM-interaction-induced magnon-phonon coupling in $Fe_2Mo_3O_8$ leads to the formation of topological magnon polarons in the magnon-phonon resonant region, with prominent features that are observed by our neutron spectroscopy measurements. Additionally, it gives rise to other intriguing phenomena even away from the anticrossing region, such as the anomalous phonons carrying spins (Fig. 3 and Supplementary Fig. 2), low-energy excitations with electric dipole activity[30,31], and the emergence of thermal Hall effect[33]. These results showing strong magnon-phonon coupling and hybrid excitations in $Fe_2Mo_3O_8$ suggest it to be a prime candidate in developing phonon-controllable spintronic devices.

## Methods

### Single-crystal growth and characterizations
High-quality single crystals of $Fe_2Mo_3O_8$ were grown by the chemical-vapor-transport method[44,45]. The well mixed raw powder materials of

$Fe_2O_3$, $MoO_2$ and Fe in a stoichiometric molar ratio of 2:9:2 were sealed in an evacuated quartz tube with $TeCl_4$ as the transport agent. The tube was placed into a two-zone horizontal tube furnace with the hot and cold end set at 980 °C and 845 °C, respectively. These temperatures were maintained for 10 days for the crystal growth, after which they were cooled naturally in the furnace to room temperature. The magnetization was measured with a 15.1-mg single crystal using the vibrating sample magnetometer option integrated in a Physical Property Measurement System (PPMS-9T) from Quantum Design.

### Neutron scattering experiments
Our neutron scattering measurements were performed on 4SEASONS, a time-of-flight spectrometer located at the MLF of J-PARC in Japan[46] and EIGER, a thermal-neutron triple-axis spectrometer located at the SINQ of PSI in Switzerland[47]. The sample array consisted of ~150 pieces of single crystals weighing about 3.39 g in total. They were coaligned using a backscattering Laue X-ray diffractometer and glued on aluminum plates using a trademarked fluoropolymer CYTOP-M. These plates were assembled together on an aluminum holder, which was then mounted into a closed-cycle refrigerator for measurements, in a manner that the $(H, 0, L)$ plane was the horizontal plane, as shown in Supplementary Fig. 1a. For the measurements on 4SEASONS, we chose a primary incident energy $E_i = 30.04$ meV and a Fermi chopper frequency of 250 Hz, with an energy resolution of 1.56 meV at the elastic line. Since 4SEASONS was operated in a multiple-$E_i$ mode[48], it had other $E_i$s of 11.93 and 17.95 meV, with respective energy resolutions of 0.56 and 0.84 meV at the elastic line. Note that on direct geometry time-of-flight spectrometers such as 4SEASONS, the energy resolution is improved as the energy transfer increases. We set the angle of the incident neutron beam direction parallel to $c$ axis to be zero. Scattering data were collected by rotating the sample about [−120] direction from 60° to 180° in a step of 1° or 2° for 6 and 100 K, respectively. We

counted 20 min for each step. In this setup, we found the data with $E_i \sim 18$ meV were overlapped with and contaminated by the elastic line from the next lower $E_i$ in the region around 15.5 meV. To eliminate this, we used a similar setup as previous measurements, but suppressed the incident neutrons with $E_i \sim 12$ meV and lower by the disk chopper in the second measurements on 4SEASONS. In the work, the results with $E_i \sim 12$ meV and $E_i \sim 30$ meV were all based on the first measurement, while those with $E_i \sim 18$ meV at 100 and 6 K were based on the first and second measurements, respectively. These data were reduced and analyzed using the software suites Utsusemi[49] and Horace[50]. For the measurements on EIGER, data were collected in the $(H, 0, L)$ scattering plane with a horizontal-focusing analyzer. We fixed the final wavevector $k_f = 2.662$ Å$^{-1}$ corresponding to an energy of 14.7 meV. We used a hexagonal structure with the refined lattice parameters $a = b = 5.773(3)$ Å and $c = 10.054(3)$ Å[28]. The wavevector $\mathbf{Q}$ was expressed as $(H, K, L)$ in the reciprocal lattice unit (r.l.u.) of $(a^*, b^*, c^*) = (4\pi/\sqrt{3}a, 4\pi/\sqrt{3}b, 2\pi/c)$. In this paper, the measured neutron scattering intensities $S(\mathbf{Q}, E)$ from 4SEASONS were corrected by the magnetic form factor of Fe$^{2+}$ ions and divided by the Bose factor via $\chi''(\mathbf{Q}, E) = |f(\mathbf{Q})|^{-2}(1 - e^{-E/k_B T})S(\mathbf{Q}, E)$, where $k_B$ was the Boltzmann constant.

## Theoretical calculations of the bands and their topology

We start the calculations with an effective two-dimensional model that contains both the magnon ($H_m$) and phonon ($H_p$) terms. The symmetry allowed $H_m$ on a bipartite honeycomb layer with two inequivalent Fe sites in Fig. 1c of the main text can be written as:

$$H_m = \sum_{i<j} J_{ij}\mathbf{S}_i \cdot \mathbf{S}_j - \sum_i \Delta_i (S_i^z)^2 + \sum_{\langle ij \rangle} \mathbf{D}_{ij} \cdot (\mathbf{S}_i \times \mathbf{S}_j). \quad (1)$$

Here, $J_{ij}$ is the Heisenberg exchange interaction between the spins $\mathbf{S}_i$ and $\mathbf{S}_j$, which is considered up to the third-nearest neighbor (TNN) to fit the magnon spectra shown in Fig. 2 of the main text. It is noted that the nearest-neighbor (NN) exchange interaction $J_1$ and TNN exchange interaction $J_3$ are homogeneous over the whole lattice while the next-nearest-neighbor (NNN) exchange interactions can take different values for the bonds between the Fe$_t$ sites (denoted by $J_2^t$) and those between the Fe$_o$ sites (denoted by $J_2^o$). $\Delta_i$ is the single-ion anisotropy constant of the local spin at the site $i$, which can be distinct for Fe$_t$ (denoted by $\Delta^t$) and Fe$_o$ (denoted by $\Delta^o$) sites as well. $\mathbf{D}_{ij}$ is the vector of the DM interaction between the NN sites $i$ and $j$. This term is present because in this case the midpoint between any NN Fe sites is no longer an inversion symmetry center[51]. Due to the preservation of the mirror symmetry with the mirror plane perpendicular to the honeycomb layer passing through any two NN Fe sites, the NN DM interaction here only has the in-plane component[51], as shown in Fig. 1c of the main text. The phonon part $H_p$ considering only Fe$^{2+}$ ions can be expressed as follows in the harmonic approximation:

$$H_p = \sum_i \frac{\mathbf{P}_i^2}{2M} + \frac{1}{2} \sum_{i,j} \mathbf{u}_i^T K_{ij}(\mathbf{R}_i^0 - \mathbf{R}_j^0)\mathbf{u}_j, \quad (2)$$

where $M$ is the ion mass of Fe$^{2+}$, $\mathbf{P}_i$ is the momentum of the ion at the site $i$, $\mathbf{u}_i$ is the displacement of the ion $i$ from its equilibrium position $\mathbf{R}_i^0$, and $K_{ij}$ is the dynamical matrix along the bond $ij$.

Magnon-phonon coupling can naturally arise in Eq. (1) when lattice vibrations are taken into account, because the exchange interaction $J_{ij}$ and the DM vector $\mathbf{D}_{ij}$ depend on the positions of the ions $i$ and $j$. In a collinear antiferromagnet with perpendicular easy-axis anisotropy, to the lowest order of $\mathbf{u}_i$ and $\delta\mathbf{S}_i = \mathbf{S}_i - \langle\mathbf{S}_i\rangle$, where $\langle\mathbf{S}_i\rangle$ is the ground state expectation value of $\mathbf{S}_i$, the isotropic Heisenberg interaction leads to a cubic magnon-phonon coupling term[15]. This term alone is only capable of causing the softening and broadening of spin waves[52]. On the other hand, the anisotropic DM interaction gives rise to a quadratic magnon-phonon coupling term[9–11], which can create a gap between the magnon

and phonon bands, leading to the formation of magnon polarons. It is also worth noting that magnetoelastic coupling, which arises from single-ion magnetostriction and is generally present in magnets with crystalline anisotropy[1,13,14], primarily couples out-of-plane phonons with in-plane magnons[1,13,14]. However, this mechanism contradicts the experimental observations where the main hybridizations occur between magnons and in-plane polarized phonons (Fig. 2a, b). Additionally, the calculated acoustic phonons with out-of-plane polarization have lower energy than the magnons, suggesting the negligible hybridization through magnetoelastic coupling (Supplementary Fig. 4). Hence, we consider the DM-interaction-induced magnon-phonon coupling as the dominant term in Fe$_2$Mo$_3$O$_8$, and for simplicity, we neglect the effects of isotropic Heisenberg interactions and magnetoelastic coupling. The final expression for this coupling $H_{mp}$ is given by,

$$H_{mp} = \frac{DS}{|\mathbf{R}_{ij}^0|} \sum_{\langle ij \rangle} \left(\mathbf{u}_i - \mathbf{u}_j\right)\left(\hat{I}_3 - \hat{\mathbf{R}}_{ij}^0 \hat{\mathbf{R}}_{ij}^0\right)\left(\delta\mathbf{S}_i + \delta\mathbf{S}_j\right), \quad (3)$$

where $D = |\mathbf{D}_{ij}|$ is the magnitude of the DM interaction, $|\mathbf{R}_{ij}^0|$ is the bond length, $S = 2$ is the total electron spin of the Fe$^{2+}$ ion, $\hat{I}_3$ is a $3 \times 3$ identity matrix, $\hat{\mathbf{R}}_{ij}^0$ is the unit vector along bond $ij$, and $\hat{\mathbf{R}}_{ij}^0 \hat{\mathbf{R}}_{ij}^0$ is the Kronecker product between two $\hat{\mathbf{R}}_{ij}^0$ s. Note that for a collinear antiferromagnet with moments aligned along $c$ axis, spins precess about the $c$ axis. In this case, only phonons involving in-plane displacements can couple to magnons according to Eq. (3), because $\delta\mathbf{S}_i$ only has the in-plane component. This can be significantly different from the magnetoelastic coupling mentioned earlier[1,13,14].

In general, the dynamical matrix $K_{ij}$ at the bond $ij$ has 6 (3) independent parameters for a three- (two-) dimensional system. For our two-dimensional effective model, it can be assumed that $K_{ij}$ only has the longitudinal components along the bond $ij$ to reasonably reduce the number of tuning parameters. Then, the elastic potential energy of the phonons can be expressed as following by the Hooke's law[11],

$$\frac{1}{2} \sum_{i<j} k_{ij} \left[\hat{\mathbf{R}}_{ij}^0 \cdot \left(\mathbf{u}_i - \mathbf{u}_j\right)\right]^2. \quad (4)$$

Here, $k_{ij}$ is the longitudinal spring constant of the bond $ij$. To fit the phonon dispersions of Fe$_2$Mo$_3$O$_8$, it is considered up to the fourth nearest neighbor (FNN), with the NN $k_1$, NNN $k_2^t$ and $k_2^o$, TNN $k_3$, and FNN $k_4$.

By using the standard Holstein–Primakoff transformation, local spins can be mapped to canonical bosons to the leading orders: $S_i^+ = \sqrt{2S}a_i$ and $S_i^z = S - a_i^\dagger a_i$ when $i$ belongs to the Fe$_t$ sites, or $S_i^+ = \sqrt{2S}b_i^\dagger$ and $S_i^z = b_i^\dagger b_i - S$ when $i$ belongs to the Fe$_o$ sites. Then, the Hamiltonian of the coupled system can be expressed in a generalized Bogoliubov-de Gennes (BdG) form as $H = \frac{1}{2}\mathbf{X}_k^\dagger \hat{H}(\mathbf{k})\mathbf{X}_k$, where $\mathbf{X}_k = \left(a_k, b_k, a_{-k}^\dagger, b_{-k}^\dagger, \mathbf{u}_k, \mathbf{P}_k\right)$. Here, $\mathbf{u}_k(\mathbf{P}_k)$ is the four-vector for the two-dimensional displacements (momenta) of the Fe$_t$ and Fe$_o$ sites. It is noted that their out-of-plane displacements and momenta are omitted because they are decoupled from the other degrees of freedom in our simple effective model. In this representation, the commutation relation of the $\mathbf{X}_k$ vector is

$$g \equiv \left[\mathbf{X}_k^\dagger, \mathbf{X}_k\right] = \begin{bmatrix} I_2 & & & \\ & -I_2 & & \\ & & & -iI_4 \\ & & iI_4 & \end{bmatrix}. \quad (5)$$

The eigenvalues and eigenvectors of the coupled system satisfy

$$g\hat{H}(\mathbf{k})|\phi_{nk}\rangle = \sigma_{nn}E_{nk}|\phi_{nk}\rangle, \langle\phi_{nk}|g|\phi_{mk}\rangle = \sigma_{nm}, \quad (6)$$

where $\sigma = \sigma_z \otimes I_6$ acts on the particle-hole space. In the BdG formulation of the Hamiltonian of the coupled system, the second half of the

eigenstates are redundant to the first due to the artificial particle-hole symmetry. In Fig. 5 of the main text, only the lowest four independent energy bands are displayed to make comparison with the experimental results.

The Berry curvature $\Omega_{n\mathbf{k}}$ of the eigenvector $|\phi_{n\mathbf{k}}\rangle$ can be defined as

$$\Omega_{n\mathbf{k}} = i\langle \nabla_{\mathbf{k}}\phi_{n\mathbf{k}}|g^{-1} \times |\nabla_{\mathbf{k}}\phi_{n\mathbf{k}}\rangle. \tag{7}$$

Then, in the spirit of the momentum space discretization method[53], the gauge-invariant Chern number and Berry curvatures can be computed from Eq. (7).

We note that our effective two-dimensional model used to obtain the results in Fig. 5 of the main text is a simplified model, but since the magnons of this system are perfectly two dimensional, and the out-of-plane phonons do not couple to the magnons at the leading order (Eq. (3)), it is able to capture the essence of the magnon polarons (Figs. 2 and 4 in the main text). Nevertheless, to account for the three dimensionality of the phonons, we have also developed a three-dimensional model (Supplementary Fig. 4).

## Data availability
The data supporting the findings of this study are available from the corresponding author J.W. upon reasonable request.

## Code availability
The codes used for the theoretical calculations of magnon-polaron bands in this study are available from the corresponding author J.W. upon reasonable request.

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

## Acknowledgements

We would like to thank Qi Zhang, Yuan Wan, Peng Zhang and Xiangang Wan for stimulating discussions. We also thank Hongling Cai for allowing us to use their X-ray diffraction machine. The work was supported by National Key Projects for Research and Development of China with Grant No. 2021YFA1400400 (J.-X.L. and J.W.), National Natural Science Foun-dation of China with Grant Nos. 12225407, 12074174 (J.W.), 92165205 (J.-X.L.), 12074175 (S.-L.Y.), 11904170 (Z.-Y.D.), and 12004191 (W.W.), Natural Science Foundation of Jiangsu province with Grant Nos. BK20190436 (Z.-Y.D.) and BK20200738 (W.W.), China Postdoctoral Science Foundation with Grant Nos. 2022M711569 and 2022T150315, Jiangsu Province Excellent Postdoctoral Program with Grant No. 20220ZB5 (S.B.), and Fundamental Research Funds for the Central Universities. We acknowledge the neutron beam time from J-PARC with Proposal Nos. 2020B0002 and 2021I0001, and from EIGER with Pro-posal No. 20200062.

## Author contributions

J.W. conceived the project. S.B. prepared the samples with assistance from Y.S., Z.H., J.L., X.Z. and B.Z. S.B., Y.S., R.K., M.N., T.F. and Z.H. carried out the neutron scattering experiments. S.B. and J.W. analyzed the experimental data. Z.-L.G., Z.-Y.D., W.W., S.-L.Y. and J.-X.L. per-formed the theoretical analyses. J.W., S.B., Z.-L.G. and J.-X.L. wrote the paper with inputs from all co-authors.

## Competing interests

The authors declare no competing interests.
