## [Peer Review File · Nature Communications]

Reviewers' Comments:

Reviewer #1:

Remarks to the Author:

This paper uses inelastic neutron scattering to study excitations in Fe₂Mo₃O₈. The central claim of the paper is that they discovered magnon polarons that have both phonon and magnon characters never seen before. The paper is well-written, and the data is very nice. Personally, I want to support the paper for publication, but I have many questions concerning this study, and I feel that the authors should carry out new measurements and/or new analysis to clarify the results before publication, as I am not sure that the central claims are correct. The paper as written cannot be published

1. The system orders at 60 K. In previous work, the system is believed to be ferrimagnetic, but the authors claim that the system is a 2D collinear antiferromagnet. If this is the case, the authors should provide the precise magnetic structure as well as moment size from the refinements. Without this information, it is difficult to determine what is going on.

2. One of the key claims is that spin waves are around 10 and 14 meV (Fig. 1e), and they are almost dispersionless. What is the reason for such a large anisotropy gap? Why can such weak dispersion support $T_N=60$ K? From recent work on sister material Ni₂Mo₃O₈ (Nat. Comm. 14, 2051 (2023)), it was claimed that two flat modes near 10-20 meV are crystal field level excitations from the tetrahedron site. Given that Fe₂Mo₃O₈ has an identical structure to that of Ni₂Mo₃O₈, how can the two Fe sites (tetrahedral and octahedral sites) have the same moment to form a collinear AF structure as claimed? What is the ground state and CEF levels in the system?

3. Of course, I agree with the authors that vanishing intensity above T_N for the 10 and 14 meV mode would suggest that they are spin waves. But I am not sure they are spin waves from tetrahedron or octahedral sites. Are there spin wave modes below 1 meV like in Ni₂Mo₃O₈?

4. One of the most interesting data, in my opinion, is the observation in Fig. 2 and Extended Data Fig. 2. I would suggest putting both in the main text. The central claim made here is that low-energy excitations at relatively low Q are mixed phonons and magnons, and therefore they disappear above T_N , leaving only phonons at high Q surviving above T_N . While the data is very nice, I am not sure I buy this argument. Typically, magnon-phonon interaction happens near the intersection points of magnons and phonons (Ref. 22; J. Phys. C 9, 1075 (1976); Phys. Rev. 188, 786 (1969)), unless magnon broadening is induced by Debye-Waller factor (Phys. Rev. B 100, 224427 (2019); Nat. Comm. 13, 4037 (2022)). However, the data in Fig. 2 and Extended Fig. 2 show the "mixed phonon-magnon mode" around 4-5 meV, about 5 meV below the supposed magnon mode? How can this be possible? Without neutron polarization analysis to determine the nature of these excitations, I would argue that this is pure speculation. I cannot imagine that spin wave intensity can mix with phonon and be dragged from 10 meV to 4-5 meV, giving rise to the claimed mixed excitations. This is way too speculative without proof. One thing I would suggest the authors do is to test if these excitations follow the Bose population factor. Both phonons and magnons must follow Bose factor, and mixed excitations should follow the Bose factor too. Since the structure factor for acoustic phonons is precisely known at each Bragg position, this can be directly checked with data in Fig. 2a and Extended Fig. 2a to determine if the intensity variation obeys the phonon structural factor. Without neutron polarization analysis, this might shed some light on the validity of the claim. You can use data above T_N as the background, as phonons without magnetic order must follow the standard structural factor.

5. The dispersive mode in Fig. 2c and Extended Data Fig. 3 is again claimed to be a phonon-magnon mixed mode and they disappear above T_N . But here, I have a major problem: there are no spin waves below 10 meV based on the authors' argument. How can phonons acquire spin wave character at energies without spin waves and in different parts of reciprocal space? I would say that this is pure speculation without any basis. Could this be due to anisotropic lattice vibration, similar to the claims on CrGeTe (see ref. above)? Do the intensities follow the expected structural factor for the c-axis acoustic mode, which acoustic phonon mode?

I am worried that the authors may have misidentified phonons and magnons in this system due to its complexity. It is hard for me to believe that two Fe sites in totally different local environments (Tetrahedral and Octahedral) would have the same moment and behave similarly. I would have thought spin waves for these two sites would have rather different energy scales. Without careful additional analysis and/or polarized neutron scattering experiments, I am not sure that the results will stand the test of time.

Overall, I think the authors took very nice data and the results are quite interesting. I am not, however, convinced by the interpretation of these results. I strongly encourage the authors to take the above considerations into account. The current version of the paper is clearly not publishable.

Reviewer #2:

Remarks to the Author:

The manuscript presents an experimental observation of the topological magnon-polaron in a multiferroic $\text{Fe}_2\text{Mo}_3\text{O}_8$ using neutron spectroscopy. The topological property of quasi-particle bands, which includes magnons and phonons, are currently the focus of spintronics research. Despite several proposed generation mechanisms, experimental detection of the topological magnon-polaron remains a challenging task. Thus, this work is timely and will undoubtedly draw strong attention from the community. The authors provide sufficient experimental data for the magnon-phonon band hybridization and describe its topological property with a reasonable theoretical model. Furthermore, the manuscript is well-written and accessible to non-experts. I believe this work will be of high interest to the broad readership of Nature Communications.

Before recommending the publication of this manuscript in Nature Communications, the authors have to address the following points:

1. The strong magnon-phonon interaction is critical in this work. Based on the 2D model, authors argue that the magnon-phonon coupling is dominantly induced by the DM interaction because the DM interaction is larger than the exchange interaction and the exchange-induced magnon-phonon coupling is cubic interaction in the 2D model. However, the lattice vibration also disturbs the perpendicular magnetic anisotropy via the crystal field effect and usually this is a dominant contribution to the magnetoelastic interaction in conventional magnets. Additionally, this effect can open the band gap between magnon and phonon bands and generate nontrivial topology of the magnon-polaron bands [see Ref. 13 and see also Shu Zhang et al., PRL 124, 147204 (2020)]. Authors should justify why they neglected such an effect in their work.

2. In this paper, the authors state that they observed the long-sought magnon polarons. However, recently there was a report on the first experimental observation of the topological magnon-polaron in a monolayer antiferromagnet FePSe_3 by the magneto-Raman spectroscopy [J. Luo et al., Nano Lett. 23, 2023-2030, (2023)]. The authors should cite this paper and discuss it appropriately.

Reviewer #3:

Remarks to the Author:

In their manuscript "Direct observation of topological magnon polarons in a multiferroic material", the authors present neutron spectroscopy measurements on $\text{Fe}_2\text{Mo}_3\text{O}_8$ combined with calculations indicating the presence of band-inverted magnon polarons induced by the DM interaction that are topologically non-trivial. In general I found the manuscript very interesting and mostly have just a few comments/questions/suggestions. However, I do wonder whether the statement "...direct spectroscopic evidence with their delicate band structures being explicitly unveiled by neutron spectroscopy has not been available yet." might be a little too strong without further clarification. Certainly if I look at some of the references given, such as for Mn_3Ge , there does seem to be INS evidence for anticrossings and

level repulsion between phonon and magnon branches with resultant magnon-polaron excitations.

Mostly the manuscript is well written, with just a few occasions when the English could be improved for greater clarity, but I assume the editorial team will be able to assist with this. The figures are clear and of good quality. In the case of HKL-E colour plots, reference should be made to Extended Data Table 1 should be made in the main text to clarify the integration ranges over the other two dimensions.

P1 Col2 Para2 It is stated that "...more intense magnetic scattering occurs at Bragg peak (1, 0, 0) instead of (1, 0, 1) (Fig. 1d), indicating an antiferromagnetic ground state instead of the ferrimagnetic state". The strong intensity at (100) rather than (101) indicates a C-type antiferromagnetic ground state as opposed to either an A-type AF, ferrimagnetic or ferromagnetic ground state, which were the 4 possibilities for a collinear structure. I therefore think that the sentence in the text needs to be careful on this point.

P2 Fig.1

In the caption to part b I would suggest the inclusion of the space group information P63mc (#186) to state explicitly that this is a polar space group. Alternatively this information could be included in the main text, but I would like to see it stated somewhere given the importance of the spatial symmetry breaking.

Fig.1e What is the intensity at (100) at around 17meV that has been cut off by the maximum measured energy transfer.

P2 Col1 Para2 It is stated that "To examine the excitation spectra in a larger momentum-energy space with finer resolution, we next performed INS measurements on a time-of-flight spectrometer." Do you mean finer energy or Q resolution or both? While it is quite possibly the case that you had better energy resolution on 4SEASONS than EIGER, in general resolution is better in a TAS experiment than in a TOF experiment, so this could be confusing to people. I would suggest you either go with: "To examine the excitation spectra in a larger momentum-energy space with finer resolution, we next performed INS measurements on the time-of-flight spectrometer 4SEASONS." or "To examine the excitation spectra in a larger momentum-energy space, we next performed INS measurements on a time-of-flight spectrometer."

P2 Col2 When talking about low energy acoustic modes, it is stated "show up at (1, 0, 1) (Extended Data Fig. 2b), but are not observed at the intense magnetic Bragg peak (1, 0, 0) from which the assumed acoustic magnon bands if present should stem (Fig. 1d,e)". I am a little confused by this. In main Fig2c there are clear dispersing modes appearing from (100) going to a maximum at (101), while the tail on the Bragg peak makes it hard to tell whether there is a gap at (100) along the H and HH0 directions. Meanwhile looking at extended Fig2b, there is a clear gap of around 5 meV at (101), but panel a shows something similar at (100).

It is also stated that "The scattering intensities of the low energy modes become stronger as the wavevector Q increases either in or out of plane (Fig. 2a-c)". I agree that this trend is definitely seen in 2a-b, but for 2c the intensity looks greater at L small than L large, with the stats looking much noisier for L=2-4. Is this an artifact coming from the regions of reciprocal space covered which have been symmetrised over, such that there was more detector coverage for low L?

P3 Fig.2

Fig.2c What does it mean "Data in c have been folded along [001] direction to improve the statistics"? If some symmetry operations have been applied, please make it clearer what these are.

The description of the spurions seen in panels a-c should appear in the caption before starting to describe d1, e1.

P4 Fig.3

Fig.3g,i Is the asymmetry in the peak shapes for phonon to magnon conversion seen in g and i consistent with the resolution ellipsoid on 4SEASONS?

Fig.3h,j In both the main text and the figure caption it is suggested that the anticrossing point is where Δ_q has its minimum, but certainly in the case of Fig.3h the minimum looks to be shifted to slightly higher energy transfers.

P8 Neutron scattering experiments methods section

"They were coaligned and glued on aluminum plates by a backscattering Laue X-ray diffractometer." Out of interest, what kind of glue was used? Also this sentence needs to be reordered to "They were coaligned using a backscattering Laue X-ray diffractometer and glued on aluminum plates."

"Note that in the measurements, the energy resolution would be improved as the energy transfer increased." It should be clarified that this is always the case for direct geometry time-of-flight spectrometers, perhaps: "Note that on direct geometry time of flight spectrometers such as 4SEASONS, the energy resolution is improved as the energy transfer to the sample increases."

Extended Data Fig.1

Could you add some information about the cross sectional area of the sample plates in the beam, and the thickness of the stacked plates? In relation to this, did you investigate whether there was any significant effect of absorption as you rotated your sample?

ED Fig.2

The spurions seen at $H \sim 0.5$ have been commented on in the main text, but some mention should also be made in the supplementary material. In particular there look to be some kind of extended features in Fig.2b at low energy transfers in addition to isolated spots.

ED Fig.4

As in the main text, the bar marking the anticrossing region seems displaced from the minimum value of Δ_q . Is this understood?

ED Table2

Please provide error bars?

Response to the Reviewers

We sincerely thank all the reviewers for their dedicated efforts in evaluating our manuscript. We are grateful that they all found our work to be of interest and complimented its quality. Reviewer #1 described our data as “*very nice*”, and Reviewer #3 acknowledged the clarity and good quality of the figures in our work. Reviewer #2 noted that our study “*is timely and will undoubtedly attract strong attention from the scientific community*”, and recommended its publication in Nature Communications once the comments have been addressed. Reviewer #3 raised several detailed and technical questions to further enhance the clarity and precision of our statements. After addressing these issues, we believe that Reviewer #3 will also recommend our paper for publication.

Reviewer #1 expressed personal support for our paper but also raised some questions regarding our study. Reviewer #1’s main concerns were related to the collinear antiferromagnetic state in $\text{Fe}_2\text{Mo}_3\text{O}_8$ and the discussion of elastic neutron scattering results in the manuscript. We believe the reviewer misunderstood our elastic scattering results as well as the discussions on the magnetic ground state. We apologize for the confusion the original version might have caused. In the revised work, we will provide further clarification to address these concerns adequately. Regarding the presence of low-energy phonons with magnonic component, we acknowledge that this aspect needs to be explained more clearly in the revised manuscript. We appreciate the reviewer’s comment, and we will revise the manuscript to provide a better understanding of this phenomenon.

We value all the insightful comments and constructive suggestions provided by the reviewers, which have significantly contributed to the improvement of our work. We have taken these recommendations and criticisms seriously, and modified the manuscript accordingly. The revised parts are highlighted in red, and we have included a summary of the changes at the end of our point-by-point response to the reviewers. We hope these will address the reviewers' concerns, and we look forward to their recommendations for the publication of our manuscript in Nature Communications.

Response to Reviewer #1

This paper uses inelastic neutron scattering to study excitations in Fe₂Mo₃O₈. The central claim of the paper is that they discovered magnon polarons that have both phonon and magnon characters never seen before. The paper is well-written, and the data is very nice. Personally, I want to support the paper for publication, but I have many questions concerning this study, and I feel that the authors should carry out new measurements and/or new analysis to clarify the results before publication, as I am not sure that the central claims are correct. The paper as written cannot be published.

Response: We sincerely appreciate the reviewer's favorable feedback on the quality of our paper and the personal support for its publication. We are grateful for the questions and suggestions raised, which have allowed us to better elucidate our findings and further improve our work. In response to the reviewer's concerns, we have provided additional clarification regarding the magnetic structure of Fe₂Mo₃O₈ and conducted new analyses to better understand the nature of the low-energy phonons. Our point-by-point response below and the corresponding changes made in the revised manuscript address all of the reviewer's concerns and provide a more convincing argument for our central claim for the existence of topological magnon polarons in Fe₂Mo₃O₈. We hope now the reviewer will find the revised manuscript, along with our detailed responses satisfactory, and recommend it for publication.

Q1:

1. The system orders at 60 K. In previous work, the system is believed to be ferrimagnetic, but the authors claim that the system is a 2D collinear antiferromagnet. If this is the case, the authors should provide the precise magnetic structure as well as moment size from the refinements. Without this information, it is difficult to determine what is going on.

Response:

We appreciate the reviewer's comment and concern regarding the magnetic structure of Fe₂Mo₃O₈. We apologize for any confusion caused by our previous statement. We would like to clarify the magnetic ground state in Fe₂Mo₃O₈ and the terminology used in our manuscript below.

The magnetic ground state of Fe₂Mo₃O₈ has been adequately studied since 1970s (Refs. 34 and 35). It is now widely accepted that Fe₂Mo₃O₈ exhibits a long-range collinear antiferromagnetic order below $T_N \sim 60$ K, with magnetic moments aligned along the c -axis (Refs. 34 and 36). The antiparallel yet uncompensated moments on each Fe-O layer, resulting from the different moment sizes of different Fe sites, stack antiferromagnetically along the c -axis (Fig. 1b in our main text). The precise moment sizes of the different Fe sites were determined through Moessbauer measurements, with

values of $4.83 \mu_B$ for Fe_o (Fe in the octahedra) and $4.21 \mu_B$ for Fe_t (Fe in the tetrahedra) sites (Ref. 35).

Therefore, in the bulk, the ground state of $Fe_2Mo_3O_8$ is ferrimagnetic within each Fe-O layer, but the stacking fashions along the c -axis can result in either ferrimagnetic or antiferromagnetic structure. At zero field, the ground state of $Fe_2Mo_3O_8$ is antiferromagnetic; with the application of a magnetic field (Extended Data Fig. 1d) or Zn doping, a metamagnetic transition to a ferrimagnetic state occurs (Refs. 28 and 29).

We fully acknowledge the widely accepted magnetic structure of $Fe_2Mo_3O_8$ in our work. In our manuscript, when we refer to $Fe_2Mo_3O_8$ as a 2D collinear antiferromagnet, we are specifically discussing the critical exponent observed in our elastic neutron scattering, which classifies it as a 2D Ising system. This classification is consistent with the strong magnetocrystalline anisotropy along the c -axis in this material (Extended Data Fig. 1c), and the presence of flat bands along the [001] direction at relatively high energies in the magnetic excitation spectra (Fig. 2c). We apologize for any confusion caused by the terminology used, and we will provide a more detailed description of the magnetic ground state of $Fe_2Mo_3O_8$ in the revised manuscript to avoid any misconceptions.

Q2:

2. One of the key claims is that spin waves are around 10 and 14 meV (Fig. 1e), and they are almost dispersionless. What is the reason for such a large anisotropy gap? Why can such weak dispersion support $TN=60$ K? From recent work on sister material $Ni_2Mo_3O_8$ (Nat. Comm. 14, 2051 (2023)), it was claimed that two flat modes near 10-20 meV are crystal field level excitations from the tetrahedron site. Given that $Fe_2Mo_3O_8$ has an identical structure to that of $Ni_2Mo_3O_8$, how can the two Fe sites (tetrahedral and octahedral sites) have the same moment to form a collinear AF structure as claimed? What is the ground state and CEF levels in the system?

Q3:

3. Of course, I agree with the authors that vanishing intensity above TN for the 10 and 14 meV mode would suggest that they are spin waves. But I am not sure they are spin waves from tetrahedron or octahedral sites. Are there spin wave modes below 1 meV like in $Ni_2Mo_3O_8$?

Response:

We appreciate the reviewer's questions. As we understood, the essence of these two questions is whether the two modes around 10 and 14 meV are spin waves or crystal-field excitations similar to those in $Ni_2Mo_3O_8$. Therefore, we would like to address these two questions together here.

We should note that the magnetism in $Fe_2Mo_3O_8$ arises solely from the Fe^{2+} ions, while the Mo^{4+} ions form spin-singlet trimers (Refs. 28 and 29). As we explain in the response

to Q1, the magnetic unit cell in $\text{Fe}_2\text{Mo}_3\text{O}_8$ consists of four sublattices of Fe^{2+} magnetic moments, as shown in Fig. 1b of the main text. The moment sizes of Fe in the octahedral and tetrahedral sites are $4.83 \mu_B$ and $4.21 \mu_B$, respectively. We attribute the two observed modes around 10 and 14 meV to be doubly degenerate magnons originating from the two different Fe sites in the collinear antiferromagnetic state. The energy scales of these modes are roughly determined by the different single-ion anisotropy constants (Δ) at the different Fe sites, approximately following the expression of $2S\Delta$ ($S = 2$ and see Extended Data Table 2 for different Δ s at the tetrahedral and octahedral Fe sites). This conclusion is based on the following facts. First, the collapse of these two modes above the magnetic transition temperature T_N supports that they are magnons. This point was also acknowledged by the reviewer. Second, our linear-spin-wave calculations successfully reproduce these two magnon modes, both in the absence and presence of magnon-phonon coupling, as shown in Fig. 5a and b of the main text. We do not observe any spin-wave modes below 1 meV, which is also consistent with the linear-spin-wave theory, as there can only be two doubly-degenerate magnon modes in this case. Furthermore, our Raman spectra under external magnetic field (in a separate work under a third review in Nature Physics) show that there are large Zeeman splittings with Lande g factors exceeding 2 for these two modes labeled as M1 and M2 in Fig. R1a, confirming their magnonic nature. In fact, in Ref. 30, the ~ 10 meV mode was also observed by THz measurements and identified to magnons.

Regarding the large anisotropy gap observed in the spin waves, we believe it is due to the combined effect of spin-orbital coupling and crystal electric field. It varies between the octahedral and tetrahedral sites, leading to distinct single-ion anisotropy constants for these two sites. The magnetic exchange interactions in the system are relatively weak, leading to less dispersive excitations. On the other hand, the transition temperature is likely to be determined by a combination of both the exchange interactions and the spin-orbital-coupling-induced magnetic anisotropy, as discussed in a previous study [Phys. Rev. B 101, 134418 (2020)].

Regarding the comparison with the isostructural compound $\text{Ni}_2\text{Mo}_3\text{O}_8$, it is important to note that the energy scales in these two materials are rather different. In $\text{Ni}_2\text{Mo}_3\text{O}_8$, the ground state is a nonmagnetic singlet, while the interplay and competition between crystal-electric-field effect and magnetic exchange interactions give rise to a magnetic order with a relatively low transition temperature ($T_N \sim 5.5$ K) [Nat. Comm. 14, 2051 (2023), Ref. 38]. In contrast, $\text{Fe}_2\text{Mo}_3\text{O}_8$ exhibits a magnetic ground state with $T_N \sim 60$ K. As a consequence, in $\text{Ni}_2\text{Mo}_3\text{O}_8$, spin waves are observed below 1.5 meV, which correspond to the two doubly degenerate modes within the framework of linear spin-wave theory. On the other hand, the additional high-energy excitations were believed to arise from the crystal-electric-field excitations of the single ions, as evidenced by their robust temperature dependence. The energy scale of these excitations is determined by the first excited doublet of the tetrahedral Ni site [Nat. Comm. 14, 2051 (2023), Ref. 38].

As for the crystal electric field excitations, our data collected with $E_i = 60$ meV show that there is no additional mode up to 50 meV in $\text{Fe}_2\text{Mo}_3\text{O}_8$. This indicates that the crystal-electric-field energy level of the first excited state in $\text{Fe}_2\text{Mo}_3\text{O}_8$ is much higher than that in $\text{Ni}_2\text{Mo}_3\text{O}_8$.

Based on these results, these two modes can be identified to be spin waves unambiguously. We will elaborate these points in more details, cite the $\text{Ni}_2\text{Mo}_3\text{O}_8$ paper and provide appropriate discussions in the revised manuscript.

Fig. R1. **a**, Raman spectra of $\text{Fe}_2\text{Mo}_3\text{O}_8$ from 20 K to 100 K. **b**, Magneto-Raman spectra of $\text{Fe}_2\text{Mo}_3\text{O}_8$ from -9 to 9 T at 20 K, measured under linearly polarized excitation and unpolarized detection (upper panel), and right(left)-handed excitations and left(right)-handed excitations (RL and LR). **c**, Schematic atomic displacements of the P1 phonons, illustrating a pair of chiral phonons with opposite cyclotron motion of Fe ions, represented as left- and right-handed modes. **d**, The peak positions of Zeeman-split P1 phonons with linear fitting.

Q4:

4. One of the most interesting data, in my opinion, is the observation in Fig. 2 and Extended Data Fig. 2. I would suggest putting both in the main text. The central claim made here is that low-energy excitations at relatively low Q are mixed phonons and magnons, and therefore they disappear above TN, leaving only phonons at high Q surviving above TN. While the data is very nice, I am not sure I buy this argument. Typically, magnon-phonon interaction happens near the intersection points of magnons and phonons (Ref. 22; J. Phys. C 9, 1075 (1976); Phys. Rev. 188, 786 (1969)), unless magnon broadening is induced by Debye-Waller factor (Phys. Rev. B 100, 224427 (2019); Nat. Comm. 13, 4037 (2022)). However, the data in Fig. 2 and Extended Fig. 2 show the “mixed phonon-magnon mode” around 4-5 meV, about 5 meV below the supposed magnon mode? How can this be possible? Without neutron polarization analysis to determine the nature of these excitations, I would argue that this is pure speculation. I cannot imagine that spin wave intensity can mix with phonon and be

dragged from 10 meV to 4-5 meV, giving rise to the claimed mixed excitations. This is way too speculative without proof. One thing I would suggest the authors do is to test if these excitations follow the Bose population factor. Both phonons and magnons must follow Bose factor, and mixed excitations should follow the Bose factor too. Since the structure factor for acoustic phonons is precisely known at each Bragg position, this can be directly checked with data in Fig. 2a and Extended Fig. 2a to determine if the intensity variation obeys the phonon structural factor. Without neutron polarization analysis, this might shed some light on the validity of the claim. You can use data above TN as the background, as phonons without magnetic order must follow the standard structural factor.

Response:

We appreciate the reviewer's comments and agree that both the data for magnon polarons (Fig. 2) and anomalous phonons (Extended Data Fig. 2) should be included in the main text. We understand the reviewer's skepticism regarding the nature of the low-energy modes below 10 meV, as it is not immediately apparent that phonons can acquire a magnon component in the off-resonant region. It is important to address any potential misidentification of phonons and magnons in $\text{Fe}_2\text{Mo}_3\text{O}_8$ due to the complexity of the material, which is also related to Q5 and Q6 raised below. To address these concerns, below we will present several facts supporting our interpretation that the low-energy modes are phonons endowed with a magnon component. The possible mechanism and additional Raman measurements which will be presented in a separate work in collaboration with our colleagues will also be discussed.

Fig. R2. Additional INS results along [100] direction. **a-c**, Excitation spectra at $T = 6$ K, measured with $E_i = 30$ meV (**a**), $E_i = 18$ meV (**b**) and $E_i = 12$ meV (**c**) on 4SEASONS. **d-f**, Same as in **a-c**, but measured at $T = 100$ K.

(1) The two intense excitations at higher energies in Fig. 2 of main text correspond to the magnons in $\text{Fe}_2\text{Mo}_3\text{O}_8$. As we explained in response to Q2 and Q3, these two modes around 10 and 14 meV are identified as two doubly degenerate magnons, which align with the expectations of linear-spin-wave theory. However, the existence of additional low-energy modes introduces a number of bands that surpass what can be explained by linear-spin-wave theory alone. Even when accounting for the nonlinearity of spin waves, the extended spin-wave theory fails to account for the presence of multiple bands at significantly lower energies compared to linear spin-wave excitations [PRL 112, 127205 (2014), Ref. 40], as the higher-order spin excitations normally occur at higher energies. Therefore, it is reasonable to consider that these low-energy excitations could be other elementary excitations in ordered magnets, such as phonons.

Fig. R3. Low-energy phonon excitation spectra. **a-c**, INS results of the excitation spectra measured at $T = 6$ K along $[100]$ (**a**) and $[001]$ (**b,c**) directions, respectively. The solid and dashed squares in **a** mark the saddle point of phonon spectra around 5 meV at $H=1$ and 2, respectively. The spectra in **b, c** correspond to the out-of-plane variations with different H s denoted by different squares compared to those in **a**. **d-e**, Same as in **a-c**, but measured at $T = 100$ K.

(2) The scattering intensities of the low-energy excitations become stronger as the wave vector \mathbf{Q} increases. In order to highlight the comparison between magnons and phonons, we have made corrections to both the Bose factor and magnetic form factor of Fe^{2+} ions in the excitation spectra along $[100]$ direction, as shown in Fig. R2. It is evident from the corrected data that the intensities of the two intense magnon bands at higher energies remain relatively unchanged as \mathbf{Q} increases. In contrast, the intensities of the low-energy excitations become stronger, which is a characteristic feature typically associated with phonons.

(3) Another piece of evidence supporting the interpretation of the low-energy modes as a phononic origin is the behavior of a specific mode with an onset energy around 5 meV along the [100] direction. This mode is observed at the (1, 0, 1) position but is not visible at (1, 0, 0) in Figs. R3a and R3b. By examining the elastic scan in the inset of the Fig. 1d in the main text, it becomes apparent that the collinear antiferromagnetic state leads to the (1, 0, 0) peak being much more intense as a magnetic Bragg peak compared to (1, 0, 1). This finding suggests that if the mode were gapped spin waves, it would originate from (1, 0, 0) instead. Furthermore, the energy scans in Fig. R4 reveal that the excitations around 5 meV become stronger at larger Q , indicating a phononic origin. By combining the dispersions along the [100] and [001] directions, we propose that the onset excitations around 5 meV correspond to a saddle point in the phonon spectra, representing the minimum and maximum of the excitations along the [100] and [001] directions, respectively.

Fig. R4. Excitation mode around the saddle point of phonons. Upper panel shows the Constant- q cuts at (1, 0, 1) and (2, 0, 4) measured at 6 K. Lower panel is same as in upper panel but measured at 100 K.

(4) In addition to the experimental observations, our theoretical calculations provide further support for the interpretation of the low-energy modes as phonons. Our calculations considering the spin correlations and lattice vibrations between Fe^{2+} ions are depicted in Fig. 5a,b of the main text. These results successfully reproduce the two high-energy magnon bands and the low-energy acoustic phonon bands, as shown in Fig. 5a. Moreover, by considering the interactions between these magnons and phonons, our calculations accurately reproduce our central observation of anticrossings and the emergence of the topological magnon polaron, as depicted in Fig. 5b. This demonstrates the capability of our model to capture the underlying physics in $\text{Fe}_2\text{Mo}_3\text{O}_8$. To account for the three-dimensional nature of the phonons, we have also developed a three-dimensional model, as presented in Extended Data Fig. 4. Remarkably, with the increased complexity and refinement of this model, we have achieved a significantly improved agreement between the experimental results and the theoretical calculations.

Multiple phonon bands, including the saddle point and the anticrossings, can be accurately reproduced.

(5) The evidence presented in the previous points strongly suggests that the additional bands at low energies are of phononic origin. However, these bands exhibit some anomalous behaviors: above T_N at 100 K, they become invisible at small \mathbf{Q} but persist at large \mathbf{Q} , as shown in Figs. R2d-f and R3d-f. To further illustrate it, we focus on the energy scans around the saddle point at approximately 5 meV at two different \mathbf{Q} s in Fig. R4 after Bose population factor correction. It is observed that although excitations exist at both positions at 6 K, only the peak at the high- \mathbf{Q} position (2, 0, 4) is visible at 100 K. Considering that the most significant difference between these two temperatures is the establishment of long-range magnetic order, it is reasonable to speculate that the strong magnon-phonon interaction plays a crucial role in this anomalous temperature dependence (Refs. 27-33). We propose that these low-energy modes acquire some spin components through the strong magnon-phonon coupling, leading to additional magnetic scattering intensities at small \mathbf{Q} s at 6 K. On the other hand, at 100 K, as magnons collapse, phonons recover their original properties and can only be observed at large \mathbf{Q} s. The disappearance of these modes at small \mathbf{Q} s at 100 K can be attributed to the small intrinsic structure factor for phonons.

(6) The reason for this phenomenon can be that the phonons involved with magnon conversion can carry spins. The mechanism in CrGeTe_3 [Nat. Comm. 13, 4037 (2022), Ref. 52] cannot be applied to $\text{Fe}_2\text{Mo}_3\text{O}_8$. In CrGeTe_3 , only magnon bands are involved at low energies, and the spin-lattice coupling acts as a higher-order term that renormalizes the spin waves, resulting in their softening and broadening. On the other hand, in $\text{Fe}_2\text{Mo}_3\text{O}_8$, both phonon and magnon bands occur simultaneously within the same energy-momentum window and interact with each other, forming anticrossings. A relevant study on the ability of phonons to carry spin was conducted in a YIG film under a non-uniform magnetic field in Ref. 17 [Nat. Phys. 14, 500 (2018)]. The authors employed wavevector-resolved Brillouin light-scattering measurements to investigate the excitations generated by continuous-wave microwave driving with a fixed frequency. By introducing and changing a uniform magnetic field superimposed on the non-uniform magnetic field, they were able to detect the excitation signal near or away from the anticrossing (Fig. 4 in Ref. 17). Their results revealed that the light scattered by phonons away from the anticrossing exhibited circular polarization, similar to that of the magnons (Fig. 5 in Ref. 17). This observation suggests that the phonons created through the conversion of magnons indeed carry spins. We propose that the situation in $\text{Fe}_2\text{Mo}_3\text{O}_8$ is similar to that described in Ref. 17. In $\text{Fe}_2\text{Mo}_3\text{O}_8$, the dispersive phonons transform into magnons with enhanced intensity in the high-energy magnon-phonon hybridization region, as demonstrated in the Figs. 2 and 4 of our main text. This phonon-magnon conversion leads to low-energy phonons carrying spin. In fact, the observation of magnon polarons at high energies and anomalous phonons at low energies can be regarded as different manifestations of the strong magnon-phonon interaction in the resonant and off-resonant regions, respectively.

(7) As a quick reference for the reviewer, we have included the results of Raman spectra in our response as shown in Fig. R1. In the Raman spectra, we observe the saddle point of phonons at the zone center, which was discussed earlier, and we label it as P1 mode. It is worth noting that the P1 mode appears to be robust against temperature (Fig. R1a), further supporting its phononic origin. When an external magnetic field is applied, we observe a linear splitting of the two doubly degenerate modes at the zone center, labeled as M1 and M2. The slopes of the splitting are determined by the different g -factors on the octahedral and tetrahedral sites, with values of 2.4 and 2.0, respectively (Fig. R1b). In the cross-circular channels, the P1 mode also exhibits a small but noticeable splitting with a g -factor of 0.11, strongly suggesting the presence of magnetic moments (Fig. R1b and R1d). This observation implies that the P1 mode represents a pair of chiral phonons with similar circular polarization as the M1 and M2 modes, corresponding to the left- and right-handed cyclotron motion of Fe ions at the tetrahedral sites (Figs. R1c). Since the Fe ions at the tetrahedral sites rotate in the same direction as the spin precession of the spin waves (Ref. 17), the magnons and the P1 phonons share the same symmetry in the antiferromagnetic phase, allowing for a linear coupling between them in the minimal model. This coupling imparts the phonons with a magnetic moment in the form of $(\gamma/\Delta)^2\mu_{\text{mag}}$, where γ , Δ and μ_{mag} are the coupling strength, detuning between magnons and phonons, and magnon magnetic moment, respectively. More detailed discussions on this topic are beyond the main scope of our paper and will appear in a separate work in collaboration with our colleagues, which is under a third review in Nature Physics.

In summary, the phononic origin of the low-energy excitations and the acquisition of spin components at low temperatures are consistent with all the experimental observations and theoretical analyses presented in our paper. We acknowledge that further confirmation and exploration of these phenomena are warranted, and we are actively working on parallel Raman research that will provide additional evidence to support our conjecture. In the revised manuscript, we will update the figures and revise the text to clarify the content related to the phononic origin of the low-energy modes and their interaction with magnons. Regarding the neutron polarization analysis mentioned by the reviewer, we appreciate that suggestion. We agree that neutron polarization analysis can play a similar role to the polarization-resolved magneto-Raman spectroscopy presented in Fig. R1. By employing polarized neutron scattering, it would be intriguing to investigate the evolution of spin components from the saddle point at the zone center to the anticrossing region at the zone boundary. We are pleased to inform the reviewer that we have already submitted a proposal for polarized neutron scattering experiments on $\text{Fe}_2\text{Mo}_3\text{O}_8$ and are currently awaiting beamtime allocation. Since the neutron flux for polarized neutron scattering is weak, obtaining high-quality polarized neutron data requires more coaligned single crystals, which will take a significant amount of time and effort. Therefore, while we are committed to performing polarized neutron scattering measurements and reporting our findings in the future, it is beyond the scope of the present study.

We have added Fig. R3 as Fig. 3 in the main text. We have added related discussions to elaborate these issues in more details in the revised manuscript.

Q5:

5. The dispersive mode in Fig. 2c and Extended Data Fig. 3 is again claimed to be a phonon-magnon mixed mode and they disappear above TN. But here, I have a major problem: there are no spin waves below 10 meV based on the authors' argument. How can phonons acquire spin wave character at energies without spin waves and in different parts of reciprocal space? I would say that this is pure speculation without any basis. Could this be due to anisotropic lattice vibration, similar to the claims on CrGeTe (see ref. above)? Do the intensities follow the expected structural factor for the c-axis acoustic mode, which acoustic phonon mode?

Response:

We appreciate the reviewer's question. As mentioned in our response to Q4, the maximum of the dispersive mode along the [001] direction is also the minimum along the [100] direction in Fig. R3, indicating a saddle point in the phonon spectrum around 5 meV. The dispersions along the in-plane and out-of-plane directions connected by this saddle point correspond to a vibration mode involving a pair of chiral phonons with left- and right-handed cyclotron motion of Fe ions on tetrahedral sites, as illustrated in Fig. R1c. These Fe ions rotate in the same direction as the spin precession of the spin wave, allowing them to couple linearly to the degenerate magnons from the same tetrahedral sites and acquire spin components. In Fig. R4, at 6 K, the mode with spin components, resulting from strong magnon-phonon coupling, contributes additional magnetic scattering intensities at small Q s. However, at 100 K, with the collapse of magnons, this mode reverts to its original phononic nature and can only be observed at large Q s. The disappearance of this mode at small Q s at 100 K may be due to the small structure factor for phonons. This observation aligns with the expected behavior based on the structure factor of phonons.

As we discussed in the response to Q4, the mechanism proposed for CrGeTe₃ [Nat. Comm. 13, 4037 (2022)] is not applicable to Fe₂Mo₃O₈. In CrGeTe₃, only magnon bands are involved in the low-energy excitation spectra, and the spin-lattice coupling, being a higher-order (cubic) term, leads to the softening and broadening of spin waves [Eq. (2) in Nat. Comm. 13, 4037 (2022), Ref. 52]. In contrast, Fe₂Mo₃O₈ exhibits the involvement of multiple magnon and phonon bands, and the magnon-phonon coupling is a lower-order (quadratic) term that gives rise to the anticrossings in the resonant region and imparts spin components to the phonon in the off-resonant region.

We have added more discussions in response to this point in the revised version.

Q6:

I am worried that the authors may have misidentified phonons and magnons in this

system due to its complexity. It is hard for me to believe that two Fes in totally different local environments (Tetrahedral and Octahedral) would have the same moment and behave similarly. I would have thought spin waves for these two sites would have rather different energy scales. Without careful additional analysis and/or polarized neutron scattering experiments, I am not sure that the results will stand the test of time.

Response:

We appreciate the reviewer's feedback and concerns. Based on the responses provided earlier and the revisions made in the manuscript, we believe that we have addressed the issues raised regarding the magnetic structure and the identification of different excitation modes in $\text{Fe}_2\text{Mo}_3\text{O}_8$. Now, we hope that the reviewer is fully convinced that the two modes around 10 and 14 meV are spin waves, while those at lower energies are phonons, but acquired some spin components. While we acknowledge that additional polarized neutron scattering experiments and further analysis might provide further insights into the system, we believe that the present data, analyses and conclusions stand firmly on their own, and therefore including these experiments is not necessary for the scope of this particular study. However, we appreciate the suggestion and acknowledge the potential for future investigations to advance our understanding of $\text{Fe}_2\text{Mo}_3\text{O}_8$.

Overall, I think the authors took very nice data and the results are quite interesting. I am not, however, convinced by the interpretation of these results. I strongly encourage the authors to take the above considerations into account. The current version of the paper is clearly not publishable.

Response:

We are grateful to the reviewer for the positive feedback on our data and for considering our results interesting. We sincerely appreciate the valuable questions and comments raised, as they have played a crucial role in enhancing the quality and clarity of our paper. We have diligently addressed all of the concerns and incorporated the necessary changes in the revised manuscript. With these improvements, we hope that the reviewer will find the response and modifications satisfactory, and we kindly request the reviewer's recommendation for the publication of our revised manuscript in Nature Communications.

Response to Reviewer #2

The manuscript presents an experimental observation of the topological magnon-polaron in a multiferroic $\text{Fe}_2\text{Mo}_3\text{O}_8$ using neutron spectroscopy. The topological property of quasi-particle bands, which includes magnons and phonons, are currently the focus of spintronics research. Despite several proposed generation mechanisms, experimental detection of the topological magnon-polaron remains a challenging task. Thus, this work is timely and will undoubtedly draw strong attention from the community. The authors provide sufficient experimental data for the magnon-phonon band hybridization and describe its topological property with a reasonable theoretical model. Furthermore, the manuscript is well-written and accessible to non-experts. I believe this work will be of high interest to the broad readership of Nature Communications.

Before recommending the publication of this manuscript in Nature Communications, the authors have to address the following points:

Response:

We greatly appreciate the positive and insightful appraisal of our work by the reviewer. We are grateful for the reviewer's willingness to recommend the publication of our work in Nature Communications pending the addressing of some points. We have carefully considered the feedback and have prepared a point-by-point response below. We will make the necessary changes in the revised manuscript to address these points adequately.

1. The strong magnon-phonon interaction is critical in this work. Based on the 2D model, authors argue that the magnon-phonon coupling is dominantly induced by the DM interaction because the DM interaction is larger than the exchange interaction and the exchange-induced magnon-phonon coupling is cubic interaction in the 2D model. However, the lattice vibration also disturbs the perpendicular magnetic anisotropy via the crystal field effect and usually this is a dominant contribution to the magnetoelastic interaction in conventional magnets. Additionally, this effect can open the band gap between magnon and phonon bands and generate nontrivial topology of the magnon-polaron bands [see Ref. 13 and see also Shu Zhang et al., PRL 124, 147204 (2020)]. Authors should justify why they neglected such an effect in their work.

Response:

We appreciate the reviewer for raising this question, which allows us to provide further clarification on the potential contribution of the magnetoelastic interactions induced by magnetic anisotropy in $\text{Fe}_2\text{Mo}_3\text{O}_8$.

In our work, we focused on the DM-induced magnon-phonon coupling as the dominant coupling mechanism. This choice was motivated by a previous experimental study that highlighted the significance of the in-plane DM interaction and strong lattice-spin coupling in $\text{Fe}_2\text{Mo}_3\text{O}_8$, particularly in the context of the Giant thermal Hall effect (Ref. 33). Subsequent theoretical studies in Refs. 9-11 investigated the DM-induced magnon-phonon coupling and its role in the thermal Hall effect. Notably, $\text{Fe}_2\text{Mo}_3\text{O}_8$ was proposed to be an ideal candidate to study this mechanism in Ref. 10. However, experimental realization of this mechanism had not been reported before.

When analyzing the excitation spectra of $\text{Fe}_2\text{Mo}_3\text{O}_8$ with clear anticrossings, it is natural for us to consider the DM-induced magnon-phonon coupling as the primary mechanism. The fitting results of our model are consistent with this coupling scenario as well (Fig. 5 and Extended Data Fig. 4). We believe that the DM-induced magnon-phonon coupling dominates in $\text{Fe}_2\text{Mo}_3\text{O}_8$ based on the coupling between in-plane phonons and in-plane magnons being the leading order in our study. In contrast, the magnetoelastic interaction by Kittel (also known as single-ion magnetostriction) primarily considers the coupling between lattice vibrations with single-site spins, without considering the exchange interactions with spins on other sites. In the case of $\text{Fe}_2\text{Mo}_3\text{O}_8$, which exhibits a collinear antiferromagnetic structure with perpendicular easy-axis anisotropy, such magnetoelastic interaction may primarily couple out-of-plane phonons with in-plane magnons, as discussed in Ref. 13 and by Shu Zhang *et al.* in PRL 124, 147204 (2020) (Ref. 14).

Our experimental observations, including the anticrossings between in-plane phonons and in-plane magnons, are inconsistent with the expectations of single-ion magnetostriction. From the in-plane excitation spectra in Fig. 2a,b, it is evident that hybridizations occur between magnons and phonons with in-plane polarization. These results remain intact regardless of the L integration, indicating that only in-plane polarized phonons are involved in the hybridization, since neutron scattering is not sensitive to phonons with out-of-plane polarization under these conditions. In Extended Data Fig. 4a, we also calculate the three-dimensional phonons by considering both in-plane and out-of-plane polarized phonons. The results show that one of the three acoustic bands, associated with the out-of-plane polarization, is lower in energy than the magnons, suggesting the absence of hybridization through magnetoelastic coupling. The remaining two acoustic bands with in-plane polarization are found to be hybridized with magnons through the DM-induced magnon-phonon coupling.

To summarize, we select the DM-induced magnon-phonon coupling as the leading-order coupling between in-plane phonons and in-plane magnons in $\text{Fe}_2\text{Mo}_3\text{O}_8$, rather than single-ion magnetostriction. We appreciate the Reviewer for bringing up these important considerations, and we will further clarify these points in the revised manuscript.

2. In this paper, the authors state that they observed the long-sought magnon polarons. However, recently there was a report on the first experimental observation of the topological magnon-polaron in a monolayer antiferromagnet FePSe₃ by the magneto-Raman spectroscopy [J. Luo et al., Nano Lett. 23, 2023-2030, (2023)]. The authors should cite this paper and discuss it appropriately.

Response:

We appreciate the reviewer for bringing this illuminating paper to our attention. We have now included a citation to the paper and provided a relevant discussion in the final paragraph of the revised text as “We note that a recent magneto-Raman scattering study has reported the existence of topological magnon polarons due to the zigzag antiferromagnetic order in the monolayer FePSe₃, where the magnon-phonon coupling originates from the anisotropic exchange interactions. It implies that the topological nature of band-inverted magnon polarons can be inherent and resilient, irrespective of the particular form of magnon-phonon coupling.”

Response to Reviewer #3

In their manuscript "Direct observation of topological magnon polarons in a multiferroic material", the authors present neutron spectroscopy measurements on Fe₂Mo₃O₈ combined with calculations indicating the presence of band-inverted magnon polarons induced by the DM interaction that are topologically non-trivial. In general I found the manuscript very interesting and mostly have just a few comments/questions/suggestions. However, I do wonder whether the statement "...direct spectroscopic evidence with their delicate band structures being explicitly unveiled by neutron spectroscopy has not been available yet." might be a little too strong without further clarification. Certainly if I look at some of the references given, such as for Mn₃Ge, there does seem to be INS evidence for anticrossings and level repulsion between phonon and magnon branches with resultant magnon-polaron excitations.

Mostly the manuscript is well written, with just a few occasions when the English could be improved for greater clarity, but I assume the editorial team will be able to assist with this. The figures are clear and of good quality. In the case of HKL-E colour plots, reference should be made to Extended Data Table 1 should be made in the main text to clarify the integration ranges over the other two dimensions.

Response:

We appreciate the thorough review conducted by the reviewer and the positive assessment of our work, including the quality of our figures and the overall manuscript. We are grateful for the reviewer's willingness to recommend our paper for publication after addressing the comments and suggestions.

We thank the Reviewer for the suggestion to modify the sentence "has not been available yet" in our previous statement. Upon reflection, we agree that it may be too strong, and we will revise it to convey that the evidence is still rare. Regarding the association of Extended Data Tables and Figures occurring in the main text, we will ensure proper referencing or inclusion of hyperlinks in the final editable version to improve clarity and accessibility for readers. Because of the clear organization of the questions and comments by the reviewer below, we will address each of them in a point-by-point response, and make appropriate changes to the revised manuscript accordingly.

P1 Col2 Para2

It is stated that "...more intense magnetic scattering occurs at Bragg peak (1, 0, 0) instead of (1, 0, 1) (Fig. 1d), indicating an antiferromagnetic ground state instead of the ferrimagnetic state". The strong intensity at (100) rather than (101) indicates a C-type antiferromagnetic ground state as opposed to either an A-type AF, ferrimagnetic or

ferromagnetic ground state, which were the 4 possibilities for a collinear structure. I therefore think that the sentence in the text needs to be careful on this point.

Response:

We appreciate the reviewer’s suggestion. As mentioned in our response to Q1 from Reviewer #1 and the revised manuscript, we have provided additional descriptions of the magnetic structure in $\text{Fe}_2\text{Mo}_3\text{O}_8$. In the ground state of $\text{Fe}_2\text{Mo}_3\text{O}_8$, the antiparallel yet uncompensated moments on each Fe-O layer, due to the different moment sizes of different Fe sites, stack antiferromagnetically along the c -axis, as shown in Fig. 1b of our main text. This antiferromagnetic state has a net magnetic moment of zero. As a result, the $(1, 0, 0)$ peak appears more intense than the $(1, 0, 1)$ peak in the magnetic scattering (Fig. 1d). In contrast, when each ferrimagnetic Fe-O layer stacks ferromagnetically along the c -axis, induced by factors such as a magnetic field or Zn doping, the net magnetic moment is non-zero, resulting in a ferrimagnetic state. In this case, the $(1, 0, 1)$ peak becomes more intense than the $(1, 0, 0)$ peak. This intensity change between the two magnetic states is consistent with the findings of a previous neutron powder diffraction experiment (Ref. 34) and our neutron elastic scattering on a single crystal of a 25% Zn-doped $\text{Fe}_{1.75}\text{Zn}_{0.25}\text{Mo}_3\text{O}_8$ sample (Fig. R5, to appear in a separate work), which undergoes a transition to a ferrimagnetic state below a certain temperature. We will make the explanation regarding the elastic scan and magnetic states clearer in the revised manuscript.

Fig. R5. Temperature dependence of the integrated intensities of the magnetic Bragg peak $(1, 0, 0)$ for $\text{Fe}_2\text{Mo}_3\text{O}_8$ (a) and $(1, 0, 1)$ for $\text{Fe}_{1.75}\text{Zn}_{0.25}\text{Mo}_3\text{O}_8$ (b). Insets show elastic scans across $(1, 0, L)$ along $[001]$ direction at temperatures below and above the transition temperature.

P2 Fig.1

In the caption to part b I would suggest the inclusion of the space group information $P63mc$ (#186) to state explicitly that this is a polar space group. Alternatively this information could be included in the main text, but I would like to see it stated somewhere given the importance of the spatial symmetry breaking. Fig.1e What is the intensity at (100) at around 17meV that has been cut off by the maximum measured energy transfer.

Response:

We appreciate the valuable suggestions provided by the reviewer. In the revised version of the manuscript, we have included the space group information explicitly in both the main text and the caption of Fig. 1b, highlighting the polar space group and emphasizing the spatial symmetry breaking.

Regarding Fig. 1e, the intensity observed at (1, 0, 0) around 17 meV, which appears to be cut off by the maximum measured energy transfer, is a false signal caused by the small scattering angle. This phenomenon arises due to the finite width and collimation of the neutron flux. When the scattering angle is too small, more neutrons that are not scattered by the sample can directly enter the detector, resulting in a false signal. We have provided an explanation for this signal in the caption of Fig. 1e in the revised manuscript.

P2 Col1 Para2

It is stated that "To examine the excitation spectra in a larger momentum-energy space with finer resolution, we next performed INS measurements on a time-of-flight spectrometer." Do you mean finer energy or Q resolution or both? While it is quite possibly the case that you had better energy resolution on 4SEASONS than EIGER, in general resolution is better in a TAS experiment than in a TOF experiment, so this could be confusing to people. I would suggest you either go with: "To examine the excitation spectra in a larger momentum-energy space with finer resolution, we next performed INS measurements on the time-of-flight spectrometer 4SEASONS." or "To examine the excitation spectra in a larger momentum-energy space, we next performed INS measurements on a time-of-flight spectrometer."

Response:

We appreciate the Reviewer's clarification regarding the resolution in different types of spectrometers. We agree with the Reviewer's suggestion to remove the words "finer resolution" in the revised manuscript to avoid potential confusion among readers. In the revised version, we will modify the sentence to convey the information more accurately.

P2 Col2

When talking about low energy acoustic modes, it is stated "show up at (1, 0, 1) (Extended Data Fig. 2b), but are not observed at the intense magnetic Bragg peak (1, 0, 0) from which the assumed acoustic magnon bands if present should stem (Fig. 1d,e)". I am a little confused by this. In main Fig2c there are clear dispersing modes appearing from (100) going to a maximum at (101), while the tail on the Bragg peak makes it hard to tell whether there is a gap at (100) along the H and HH0 directions. Meanwhile looking at extended Fig2b, there is a clear gap of around 5 meV at (101), but panel a shows something similar at (100).

It is also stated that "The scattering intensities of the low energy modes become stronger as the wavevector Q increases either in or out of plane (Fig. 2a-c)". I agree that this trend is definitely seen in 2a-b, but for 2c the intensity looks greater at L small than L

large, with the stats looking much noisier for $L=2-4$. Is this an artifact coming from the regions of reciprocal space covered which have been symmetrised over, such that there was more detector coverage for low L ?

Response:

We appreciate the reviewer's questions and concerns regarding the low-energy acoustic modes and the intensity trend in Fig. 2. Regarding the low-energy modes, our intention was to demonstrate that they originate from phononic excitations rather than magnons. We meant to show that the excitations around 5 meV are not coming from the intense magnetic Bragg peak (1, 0, 0) but rather from (1, 0, 1), as depicted in Fig. 3a,b. The similarity observed at (1, 0, 0) in panel a of the same figure is due to the L integration over $[-3, 3]$. The excitations around 5 meV at (1, 0, 1) connect the maximum of the dispersive mode along the [001] direction and the minimum of the mode along the [100] direction. This configuration makes the point around 5 meV a saddle point in the phonon spectra. To avoid confusion, we will use the term "saddle point" instead of "gap" in the revised manuscript. We have added Fig. 3 in the main text to show more details on the low-energy acoustic phonons, and included more details characterizations on these modes, in response to this comment as well as reviewer #1's Q4 and Q5. With these, this issue should be clear.

Regarding the intensity trend in Fig. 2c, as suspected by the reviewer, the noisier statistics for $L = 2-4$ compared to 0-2 in Fig. 2c of the main text is indeed due to an artifact caused by the symmetry operation. Our scattering data were collected by rotating the sample about the $[-120]$ direction from 60 degrees to 180 degrees. As a result, the combined Horace scans mainly cover the first quadrant of the $(H, 0, L)$ plane. The unsymmetrized raw data in Fig. 2c contains the excitation spectra from $L = -2$ to 4. The symmetry operation from $-L$ to L improves the statistics only for $L = 0$ to 2. We have added proper note in the caption.

P3 Fig.2

Fig.2c What does it mean "Data in c have been folded along [001] direction to improve the statistics"? If some symmetry operations have been applied, please make it clearer what these are.

The description of the spurions seen in panels a-c should appear in the caption before starting to describe d1, e1.

Response:

We appreciate the suggestions provided by the reviewer. In response to the feedback, we will make the following changes in the revised manuscript: (i) In Fig. 2c, we will modify the description to "In c, raw data on the negative L side have been symmetrised to the positive side to improve the statistics." (ii) We will relocate the description of the spurions in panels a-c to the caption before starting the description of d1 and e1.

P4 Fig.3

Fig.3g,i Is the asymmetry in the peak shapes for phonon to magnon conversion seen in g and i consistent with the resolution ellipsoid on 4SEASONS?

Fig.3h,j In both the main text and the figure caption it is suggested that the anticrossing point is where Δ_q has its minimum, but certainly in the case of Fig.3h the minimum looks to be shifted to slightly higher energy transfers.

Response:

We appreciate the reviewer's questions. Regarding Figs. 3g,i (Figs. 4g,i now), the asymmetry in the peak shapes for phonon to magnon conversion is indeed consistent with the resolution ellipsoid on 4SEASONS. Although there is not a mature code or software to directly calculate the resolution ellipsoid convoluted into the measured data on 4SEASONS yet, we have discussed with the instrument scientists and gained qualitative insights into the effects of resolution ellipsoids. The asymmetry in peak shapes can be attributed to the "focusing" and "defocusing" conditions caused by the resolution ellipsoids. The slopes of the dispersed bands in the higher anticrossing region (Fig. 4g) are more comparable to those of the resolution ellipsoid, leading to a more pronounced difference between the "focusing" condition for $H > 1$ and the "defocusing" condition for $H < 1$, as explained later. Before discussing the data, we would like to mention that the resolution ellipsoid generally elongates along the scan trajectory in the \mathbf{Q} - E plane and perpendicular to the scan trajectory in the constant- E plane on 4SEASONS. With this in mind, the instrument scientists used Utsusemi software to calculate the scan trajectory in the \mathbf{Q} - E plane with the same experimental setup, which allowed us to deduce the shapes of the resolution ellipsoids. In Figs R6a, b, d, and e, we provide schematic ellipsoids in the simulated \mathbf{Q} - E plane, revealing different conditions for $L = 2, 3$ (Figs. R6a, b) and $L = -2, -3$ (Figs. R6d, e). The resolution ellipsoids mostly have a positive slope for $L = 2, 3$, but it has both positive and negative slopes centered around $H = 1$ for $L = -2, -3$.

To illustrate the influence of the resolution ellipsoids, let's consider the example of constant- E scans at 14.3 meV (Fig. 4g2). By narrowing the integration range to $2 < L < 3$, where only the positive resolution ellipsoid is relevant, we find that the dispersion for $H > 1$ has a positive slope matching that of the resolution ellipsoid. Consequently, the intensities are dominated by the "focusing" condition, resulting in a peaked shape in the intensity profile. Conversely, the dispersion for $H < 1$ exhibits a negative slope, leading to the "defocusing" condition and a broadened peak shape (Fig. R6c). In the case of $-3 < L < -2$, where the slopes of the resolution ellipsoids are negative and positive for $H > 1$ and $H < 1$, respectively, both sides correspond to the "defocusing" condition. Consequently, broad peaks are observed for both regions (Fig. R6f). The final profiles of the constant- E scans shown in the main text include features with different L values, ranging from -3 to 3. While the asymmetry affects the full width at half maximum (FWHM) of the peaks, the integrated intensities remain comparable. In Fig. 4h,j, we averaged the integrated intensities and peak centers for both $H > 1$ and $H < 1$ sides, ensuring that the asymmetry does not significantly impact the conclusions

drawn regarding magnon polarons. We have updated the caption accordingly to reflect the effect of the resolution ellipsoid on the peak asymmetry.

We have noticed that the shift mentioned by the reviewer particularly occurs in the higher anticrossing region around 14.3 meV in Fig. 4h. We believe that the main reason for this shift is related to the different band structures in that energy range. In Fig. 2d, it can be observed that the uncoupled magnon and phonon bands exhibit similar group velocities as they disperse towards the band top and come close to touching each other around 14.3 meV. This is in contrast to the situation around 11.3 meV, where the relatively flat magnon band intersects steep phonon bands. As we move away from the anticrossing region towards higher energies, we find that the change in Δq is relatively slight for the former case in Fig. 4h but becomes more pronounced for the latter case in Fig. 4j. Additionally, significant changes in the band structures occur within a small energy window of only 0.6 meV for the anticrossing regions. There might be some fitting errors, particularly given that the integration thickness for the constant- E cuts in Fig. 4g,i is 0.1 meV, which is close to the experimental resolution limit. These factors could contribute to the slight shift observed in the data. Lastly, we acknowledge that the guidelines in panels h and j may exacerbate the observed shift. Optimizing the guidelines is currently the best approach we can take. We have made adjustments to the guidelines in the revised figures to mitigate this effect.

Fig. R6. Qualitative analyses of the resolution effects. **a,b**, Calculated scan trajectory in the \mathbf{Q} - E plane with the same experimental setup for $L=2$ (**a**) and 3 (**b**). The elongated ellipsoids are schematic representations of the resolution function shape. **c**, The constant- E scan at 14.3 meV with L integrated over $[2, 3]$. **d,e**, Same as in **a,b**, but with $L=-2$ (**c**) and -3 (**d**). **f**, Same as in **c** but with L integrated over $[-3, -2]$.

P8 Neutron scattering experiments methods section

"They were coaligned and glued on aluminum plates by a backscattering Laue X-ray diffractometer." Out of interest, what kind of glue was used? Also this sentence needs

to be reordered to "They were coaligned using a backscattering Laue X-ray diffractometer and glued on aluminum plates."

"Note that in the measurements, the energy resolution would be improved as the energy transfer increased." It should be clarified that this is always the case for direct geometry time-of-flight spectrometers, perhaps: "Note that on direct geometry time of flight spectrometers such as 4SEASONS, the energy resolution is improved as the energy transfer to the sample increases."

Response:

We appreciate the reviewer's suggestions. The glue used is CYTOP-M, a trademarked fluoropolymer. We have added this information in the revised text. We have followed the reviewer's recommendations and made other changes in the revised manuscript.

Extended Data Fig.1

Could you add some information about the cross sectional area of the sample plates in the beam, and the thickness of the stacked plates? In relation to this, did you investigate whether there was any significant effect of absorption as you rotated your sample?

Response:

We appreciate the reviewer's suggestion. The aluminum plates used in our experiment have dimensions of $26 * 26 * 0.3 \text{ mm}^3$. The crystals were glued onto four plates. Each pair of plates was stuck back to back, and the two pairs of plates were assembled with aluminum nuts, resulting in a stacking thickness of 6.4 mm. The horizontal plane during the experiment corresponds to the $(H, 0, L)$ plane, as shown in Extended Data Fig.1.

Regarding the effect of absorption, we rotated the sample about the $[-120]$ direction from $\Psi=60$ degrees to 180 degrees, where $\Psi=0$ degree is defined when the beam is perpendicular to the plates, along the $[001]$ direction. As the sample was rotated, the cross-sectional area exposed to the neutron beam varied, potentially leading to different neutron absorption. However, we believe that the effect of absorption is not significant for our results. The maximum absorption occurs when the plates are parallel to the incoming beam, corresponding to $\Psi=90$ degrees in our experimental setup. If absorption were strong, we would expect to see a dark curve along the scan trajectory at $\Psi=90$ degrees. However, upon examining the constant- E contours at $E = 0$ and other energies, we did not observe such a dark curve in the contour maps, suggesting a negligible effect of absorption due to sample rotation. Furthermore, even if a dark curve were visible, we believe it would be smeared out during the integration over L when plotting the data. We have included comments addressing these issues in the revised caption accordingly.

ED Fig.2

The spurious seen at $H \sim 0.5$ have been commented on in the main text, but some mention should also be made in the supplementary material. In particular there look to be some kind of extended features in Fig.2b at low energy transfers in addition to

isolated spots.

Response:

We appreciate the reviewer's suggestion. We have included a comment on the spurions in the revised version of the Extended Data file.

ED Fig.4

As in the main text, the bar marking the anticrossing region seems displaced from the minimum value of Δ_q . Is this understood?

Response:

We appreciate the reviewer's question. We think the displacement is attributed to the same reasons mentioned in the response to question P4 Fig.3. We have taken this into consideration and made adjustments to the guidelines in the revised figures.

ED Table2

Please provide error bars?

Response:

We appreciate the reviewer's suggestion. However, our theoretical calculations in this table do not involve errors.

Summary for the Changes in the Main Text

I. Changes in Figures and Captions

- (1) In the caption of Fig. 1, provide information about space group and magnetic structure of $\text{Fe}_2\text{Mo}_3\text{O}_8$; explain the origin of the cut-off bright spots.
- (2) In the caption of Fig. 2, change the description about symmetry operation; relocate the description of the spurions.
- (3) Add a new Fig. 3 to discuss the low-energy phonon excitations around the 5-meV saddle point.
- (4) Adjust the guidelines in Fig. 4h,j (original Fig. 3h,j); give the reason of the peak asymmetry in the caption.

II. Changes in Text Part

- (5) Revise the statement about "direct spectroscopic evidence" to convey that the evidence is still rare in the abstract and the second paragraph of the paper.
- (6) Provide a more detailed description of the lattice and magnetic structure of $\text{Fe}_2\text{Mo}_3\text{O}_8$ and adjust the textual logic before presenting our results.
- (7) Revise the statement about the elastic and inelastic neutron results from EIGER, and add appropriate discussions about $\text{Ni}_2\text{Mo}_3\text{O}_8$ and its difference from $\text{Fe}_2\text{Mo}_3\text{O}_8$ and $\text{Co}_2\text{Mo}_3\text{O}_8$ in "Magnons at high energies" section.
- (8) Revise the statement about low-energy phonons acquiring spin components in the sections of "Anomalous phonons at low energies" and "Formation of magnon polarons".
- (9) Add a "Discussions section" at the end of the paper; discuss the results in Nano Lett. 23, 2023–2030 (2023) (Ref. 43 now); compare DM-induced magnon-phonon coupling with the general magnetoelastic coupling.
- (10) Revise the experimental details in the experimental part of the "Methods" section.
- (11) Distinguish the DM-induced magnon-phonon coupling from the magnetoelastic coupling in detail in the theoretical part of the "Methods" section.
- (12) Add relevant references: Phys. Rev. Lett. 124, 147204 (2020) (Ref. 14); Phys. Rev. B 102, 094307 (2020) (Ref. 36); Nat. Commun. 14, 2051 (2023) (Ref. 38); Phys. Rev. Lett. 112, 127205 (2014) (Ref. 40); Nano Lett. 23, 2023–2030 (2023) (Ref. 43); Nat. Commun. 13, 4037 (2022) (Ref. 52).

Summary for the Changes in the Extended Data File

- (1) In the caption of Extended Data Fig. 1, provide information about the cross-sectional area and stacking thickness of the sample plates, and discuss the potential effect of neutron absorption.
- (2) Rearrange the original Extended Data Figs. 2 and 3, and update them as the new Extended Data Fig. 2 and the new Fig. 3 in the main text.
- (3) Adjust the guidelines in Extended Data Fig. 3c,d (original Extended Data Fig. 4c,d).
- (4) In the caption of Extended Data Fig. 4 (original Extended Data Fig. 5), add some comments to compare the DM induced magnon-phonon coupling with the magnetoelastic coupling.
- (5) Revise Extended Data Table 1 to apply to the updated figures of this new version.

Reviewers' Comments:

Reviewer #1:

Remarks to the Author:

I have carefully read through the replies to my original questions and I am quite happy with authors' replies. I also read through the revised paper, and feel that the authors have done a good job in addressing concerns of the referees. Overall, this is a nice paper and should be published in Nature Communications. Although I am still not totally convinced of the novel spin-lattice coupling proposed by the authors, I do agree with the authors that polarized measurements to be carried out by them should shed more future light on the system.

Reviewer #2:

Remarks to the Author:

The author's response and revised version of the manuscript have addressed my concerns satisfactorily. I believe this work will attract much attention from readers interested in topological magnons and phonons. I recommend the publication of this work in Nature Communications.

Response to the Reviewers

We wish to express our sincere gratitude to the reviewers for their dedicated efforts in evaluating our manuscript. We are delighted to report that we have effectively addressed all of the reviewers' concerns and queries through our revisions. In this revised version of the manuscript, no further questions or comments have been raised. This, in our view, reflects the reviewers' satisfaction with our responses and the improved clarity of the manuscript. We are particularly appreciative of the reviewers' positive recommendations for the publication of our work in Nature Communications.

Response to Reviewer #1

I have carefully read through the replies to my original questions and I am quite happy with authors' replies. I also read through the revised paper, and feel that the authors have done a good job in addressing concerns of the referees. Overall, this is a nice paper and should be published in Nature Communications. Although I am still not totally convinced of the novel spin-lattice coupling proposed by the authors, I do agree with the authors that polarized measurements to be carried out by them should shed more future light on the system.

Response: We express our sincere gratitude to the reviewer for the careful evaluation of our manuscript. We are appreciative of the reviewer's recommendation for the publication of our work in Nature Communications. We have emphasized the discovery of topological magnon polarons in our current study, and we maintain our confidence in their emergence through a novel spin-lattice coupling mechanism. To further investigate the behavior of low-energy chiral phonons in this material, we are committed to conducting additional polarized neutron scattering measurements.

Response to Reviewer #2

The author's response and revised version of the manuscript have addressed my concerns satisfactorily. I believe this work will attract much attention from readers interested in topological magnons and phonons. I recommend the publication of this work in Nature Communications.

Response: We express our sincere gratitude to the reviewer for the careful evaluation of our manuscript. We are appreciative of the reviewer's recommendation for the publication of our work in Nature Communications.